

# Vertical profile of atmospheric dimethyl sulfide in the Arctic Spring and Summer

**Roya Ghahreman[1], Ann-Lise Norman[1], Betty Croft[2], Randall V. Martin[2], Jeffrey R. Pierce[3], Julia Burkart[4], Ofelia Rempillo[1], Heiko Bozem[5], Daniel Kunkel[5], Jennie L. Thomas[6], Amir A. Aliabadi[7], Gregory R. Wentworth[4], Maurice Levasseur[8], Ralf M. Staebler[9], Sangeeta Sharma[9] and W. Richard Leaitch[9]**

[1] Department of Physics and Astronomy, University of Calgary, Calgary, Canada

[2] Department of Physics and Atmospheric Science, Dalhousie University, Halifax, Canada

[3] Department of Atmospheric Science, Colorado State University, Fort Collins, USA

[4] Department of Chemistry, University of Toronto, Toronto, Canada

[5] Institute of Atmospheric Physics, University of Mainz, Mainz, Germany

[6] Sorbonne Universités, UPMC Univ. Paris 06, Universite Versailles St-Quentin, CNRS/INSU, UMR8190, LATMOS-IPSL, Paris, France

[7] Environmental Engineering Program, School of Engineering, University of Guelph, Guelph, Canada

[8] Department of Biology, Laval University, Quebec, Canada

[9] Environment and Climate Change Canada, Toronto, Canada

Corresponding author: Ann-Lise Norman (alnorman@ucalgary.ca)

## Abstract

Vertical distributions of atmospheric dimethyl sulfide (DMS(g)) were sampled aboard the research aircraft Polar 6 near Lancaster Sound, Nunavut, Canada in July 2014 and on pan-Arctic flights in April 2015 that started from Longyearbyen, Spitzbergen, and passed through Alert and Eureka, Nunavut and Inuvik, Northwest Territories. Larger mean DMS(g) mixing ratios were present during April 2015 (campaign-mean of 116±8 pptv) compared to July 2014 (campaign-mean of 20±6 pptv). Observations in July 2014 indicated a decrease in DMS(g) mixing ratios with altitude up to about 3 km, and the largest



mixing ratios were found near the surface above ice-edge and open water, coincident with increased
particle concentrations. In contrast, DMS(g) mixing ratios sampled in April 2015 were as high as 100
pptv near 2500 m. The April campaign also exhibited uniform campaign-mean vertical profiles overall
although some profiles showed an increase with altitude.
GEOS-Chem chemical-transport model simulations indicate that Arctic seawater (north of 66°N)
contributes the majority of DMS(g) to the Arctic profiles (>90%) in July 2014 flight tracks which were
below 3000 m. More than 90% of DMS(g) in April 2015 was from Arctic seawater for measurements
below 500 m, but that declined to 60% for altitudes between 500 m and 3000 m. FLEXPART simulations
indicate that for summer 2014, the sampled air mass originated over Baffin Bay and the Canadian Arctic
Archipelago. Whereas, for springtime 2015, the air mass sampled on flights near Alert and Eureka
originated from Baffin Bay/Canadian Archipelago and from long-range transport (LRT) around the
northern tip of Greenland. Our results highlight the role of open water below the flight as the source of
DMS(g) during July 2014, and the influence of LRT of DMS(g) from further afield in the Arctic above
2500 m during April 2015.

## 1  Introduction

The Arctic has experienced rapid climate change in recent decades (IPCC, 2013). Its high climate
sensitivity distinguishes the Arctic from the rest of the world. The Arctic Ocean moderates Arctic climate
and has variable surface temperature and salinity as ice cover melts and freezes (Bourgain et al., 2013).
This ocean is an important source of atmospheric gases and particles (e.g. dimethyl sulfide, as well as sea
salt, organic and biogenic particles) (e.g. Bates et al., 1987; Andreae, 1990; Yin et al., 1990; Leck and
Bigg, 2005a, b; Barnes et al., 2006; Ayers and Cainey, 2007; Sharma et al. 2012). Aerosols affect the
climate by scattering/reflecting sunlight (direct effects), changing number/size of cloud droplets and
altering precipitation efficiency (indirect effects) (Twomey, 1974; Albrecht, 1989). The study of these
particles has been of interest for numerous researchers because of their importance in Arctic climate
change. Najafi et al. (2015) estimated that the net effect of aerosol is cooling the Arctic. However, there
are many uncertainties related to the estimation of effects and sources of aerosol particles.




Atmospheric oxidation of DMS(g) is the main source of biogenic sulfate aerosols in the Arctic (Norman
et al., 1999). DMS(aq) is produced by the breakdown of dimethylsulfonopropionate (DMSP) by oceanic
phytoplankton and bacteria DMSP-lyases (Levasseur, 2013) and transported to the atmosphere via
turbulence, diffusion and advection (Lunden et al., 2010). Sulfur compounds from atmospheric DMS(g)
oxidation are able to form new particles and condense on pre-existing aerosols in the atmosphere (Chang
et al., 2011). If sufficient condensable vapours are available, the particles may grow large enough to act
as cloud condensation nuclei (CCN) (Charlson et al., 1987). Although there are uncertainties in details of
the negative feedback of DMS(g) emissions to warming at the global scale or CLAW hypothesis
(Charlson et al., 1987), such as the air–sea exchange, atmospheric chemistry/reactions and cloud
microphysics (Quinn and Bates, 2011), DMS(g) emissions play a relatively more important role in climate
change in remote areas with low aerosol concentrations, such as in the Arctic (Carslaw et al., 2013,
Levasseur 2013, Croft et al., 2016a).
Dimethyl sulfide production and emission to the atmosphere vary seasonally. Production and emission
are particularly strong during the Arctic summer time due to high temperature, biological activity, and
the amount of ice-free surface area. Melting ice in the marginal ice zone, ice-edge and under-ice are
favourable for the production of DMSP and aqueous DMS(aq) by oceanic phytoplankton (Leck and
Persson, 1996; Matrai and Vernet, 1997; Levasseur 2013). After summer, aqueous phase DMS(aq)
concentrations decrease by about three orders of magnitude between August and October in the central
Arctic Ocean (Leck and Persson, 1996).
Dimethyl sulfide oxidation in the atmosphere occurs by the radical addition pathway (by hydroxyl radicals
OH and halogen oxides) and by the H abstraction pathway (by the nitrate radical $NO_3$, OH and halogens)
(Barnes et al., 2006; von Glasow and Crutzen, 2004). In general, the DMS(g) oxidation rate and pathway
depends on the available oxidants and temperature. The final products of DMS(g) oxidation by the
addition pathway are DMSO and MSA. MSA likely condenses onto pre-existing aerosols (von Glasow
and Crutzen, 2004). On the other hand, DMS(g) oxidation by the abstraction pathway leads to formation
of $SO_2$. More than half of $SO_2$ is removed from the atmosphere via dry and wet deposition and the
remaining $SO_2$ may form sulfuric acid ($H_2SO_4$) in the gas and aqueous phases (Pierce, et al., 2013).





Sulfuric acid formed in the gas phase is a key atmospheric nucleation component which is able to form
new particles that may grow to the size of CCN and affect climate (Kulmala, et al., 2004).
Previous measurements of DMS(g) in the Arctic atmosphere are limited to a few studies and field
campaigns at different locations (e.g. Sharma et al., 1999; Rempillo et al., 2011; Mungall et al., 2016).
The study of the vertical distribution of DMS (g) in the Arctic atmosphere is also limited to a few
observations. Ferek et al., (1995) reported the first measurements of DMS(g) vertical profiles over the
Arctic Ocean near Barrow in early summer 1990 and spring 1992. They reported low DMS(g) mixing
ratios (a few pptv) during spring and relatively high (a few tens pptv with some peaks around 100 to 300
pptv) during summer. They concluded that the Arctic Ocean is the potential source of DMS(g), and DMS
(g) ocean-atmosphere exchange is more important early summer due to sea ice melt.
Kupiszewski et al. (2013) measured atmospheric DMS(g) on board a helicopter and observed large
variability in DMS(g) mixing ratios over the central Arctic Ocean during summer. The median (mean)
values were around 7 (34) pptv near the surface < 200 m, 11 (22) pptv for altitude between 200 and 1000
m, and 4 (5) pptv above 1000 m.
Lunden et al. (2010) presented model results for the vertical distribution of DMS (g) in the Arctic (north
of 70ºN) during summer. They reported a variable vertical profile for DMS(g) concentrations above open
water, with the highest concentrations near the surface (around 115 and 365 pptv for the median and 95th
percentiles respectively) and an exponential decrease with height. In contrast, over the pack-ice, DMS(g)
concentrations were higher above the local boundary layer than at the surface. Also, Lunden et al. (2010)
showed that DMS(g) can be mixed downward by turbulence into the local boundary layer and act as a
local near-surface DMS source over the pack-ice. In addition, they compared modeling results with
measurements from the Arctic Ocean Expedition 2001 (AOE-2001, Leck et al., 2004; Tjernström et al.,
2004) and reported that DMS(g) was present above the local boundary layer in both the model and
observations.
For our study, atmospheric DMS(g) samples were collected in Tenax tubes during Polar 6 aircraft flights
in the Arctic. We compared these DMS(g) measurements to GEOS-Chem chemical transport model
simulations and conducted sensitivity simulations to examine the local versus long range transport (LRT)
DMS(g) sources for both the spring and summer. In addition, FLEXPART was applied in back trajectory



mode in order to investigate the DMS(g) source regions based on potential emission sensitivity
simulations. Field and sampling locations, as well as measurement/modeling methods are described in
section 2. Section 3 includes meteorological and DMS(g) measurement data. Section 4 presents
discussion of results, and comparison of measurement with modeling results (GEOS-Chem and
FLEXPART) are in section 5. The summary and conclusion of this study are reported in section 6.

## 2    Field description and methods

### 2.1    Measurements

#### 2.1.1    DMS

DMS(g) was collected aboard the research aircraft Polar 6 in the Arctic during July 2014 and April 2015,
as part of the NETCARE (Network on Climate and Aerosols: Addressing Key Uncertainties in Remote
Canadian Environments) project. The Polar 6 aircraft routes and sampling locations from 12 to 21 July
2014, and from 5 to 20 April 2015 are shown in Figures 1 and 2, respectively. The Polar 6 campaign was
based from Resolute Bay, Nunavut, and covered the Lancaster Sound area in July 2014. In April 2015,
the flights started from Longyearbyen, Spitzbergen, and passed through Alert and Eureka, Nunavut and
Inuvik, Northwest Territories.
Atmospheric DMS(g) was collected on cartridges packed with Tenax TA®. Mass flow was controlled at
200 mL/min, and a KI-treated 47 mm quartz Whatman filter was fitted at the intake of cartridge to remove
all oxidants. Two Teflon valves were placed before and after the Tenax tube to control the sampling
period, and Teflon tubing was used to transfer the sample from outside the aircraft to the sampler. The
samples were stored in an insulated container with a freezer pack after collection and in a freezer after the
flight. Sampling collection times were between 2 to 10 minutes in 2014 and 4 to 11 minutes in 2015.
A glass gas chromatograph (GC) inlet liner was used to pack 170±2 mg of Tenax. The Tenax packed in
glass tubes was cleaned by heating to 200°C in an oven with a constant He flow of around 15 mL/min for
5 hours. After cleaning, Tenax was injected with 50 µL of 1 ppm DMS(g) standard and stored in a freezer
at -25°C. Three Tenax tubes injected with standard DMS(g) along with one blank Tenax tube were





analyzed for each test period in the laboratory using a Hewlett Packard 5890 GC fitted with a Sievers
Model 355 sulfur chemiluminescence detector (SCD). Two DMS(g) standards in gas phase (1 and 50
ppmv) were used to calibrate the GC-SCD. Collection and analysis of samples were based on methods
described by Sharma et al. (1999) and Rempillo et al. (2011). There is ±12 pptv of uncertainty associated
with DMS(g) measurements with this method.
Additional tests were performed to determine if there was significant loss of DMS(g) over time after
collection. An experiment was performed to determine how long Tenax is able to store DMS(g) with no
significant loss of concentration. Tenax storage tests at -25°C showed that DMS losses were
approximately 5% and 15% after 10 and 20 days respectively (Figure 3). The DMS(g) mixing ratios
summarized in Table 1 are adjusted according to the result of this test.
2.1.2   Meteorological measurements
Meteorological measurements were performed by an AIMMS-20 (Aircraft Integrated Meteorological
Measurement System) instrument, manufactured by Aventech Research Inc., Barrie, Ontario, Canada.
This instrument was used to measure the three-dimensional, aircraft-relative flow vector (true air speed,
angle-of-attack, and sideslip), temperature, relative humidity, turbulence, and horizontal/vertical wind
speeds. Accuracy and resolution were 0.30 and 0.01 °C for temperature and 2.0 and 0.1 % for relative
humidity. More details of the instrument and corresponding aircraft measurements were recently
published in other studies from the same campaign (e.g. Leaitch et al., 2016, Aliabadi et al., 2016b, and
Willis et al., 2016).
**2.2   Model Description**
2.2.1   GEOS-Chem Chemical Transport Model
The GEOS-Chem chemical transport model (www.geos-chem.org) was used to interpret the vertical
profile of DMS(g).  We used GEOS-Chem version 9-02 at 2×2.5º resolution with 47 vertical layers
between the surface and 0.01 hPa. The assimilated meteorology is taken from the National Aeronautics





and Space Administration (NASA) Global Modeling and Assimilation Office (GMAO) Goddard Earth
Observing System version 5.7.2 (GEOS-FP) assimilated meteorology product, which includes both
hourly surface fields and 3-hourly 3D fields. Our simulations used 2014 and 2015 meteorology following
a 1-month spin-up prior to the simulation of July 2014 and April 2015.
The GEOS-Chem model includes a detailed oxidant-aerosol tropospheric chemistry mechanism as
originally described by Bey et al. (2001). The simulated aerosol species include sulfate-nitrate-ammonium
(Park et al. 2004; 2006), carbonaceous aerosols (Park et al, 2003; Liao et al. 2007), dust (Fairlie et al.
2007; 2010) and sea salt (Alexander et al. 2005). The sulfate-nitrate-ammonium chemistry uses the
ISORROPIA II thermodynamic model (Fountoukis et al. 2007), which partitions ammonia and nitric acid
between the gas and aerosol phases. Climatological biomass burning emissions are from the Global Fire
Emissions Dataset (GFED3). DMS(g) emissions are based on the Liss and Merlivat (1986) sea-air flux
formulation, and oceanic DMS(g) concentrations from Lana et al. (2011). In our simulations, DMS(g)
emissions occurred only in the fraction of the grid box that is covered by seawater and also free of sea
ice. Simulated DMS(g) oxidation occurs by reaction with OH and $NO_3$. The model also includes natural
and anthropogenic sources of $SO_2$ and $NH_3$ (Fisher et al. 2011).  Oxidation of $SO_2$ occurs in clouds by
reaction with $H_2O_2$ and $O_3$ and in the gas phase with OH (Alexander et al, 2009). Reaction rates and the
yields of $SO_2$ and MSA from DMS(g) oxidation are determined by DeMore et al. (1997) and Chatfield
and Crutzen (1990), respectively.
The GEOS-Chem model has been extensively applied to study the Arctic atmosphere, in regard to aerosol
acidity (Wentworth et al., 2016; Fisher et al., 2011) carbonaceous aerosol (Wang et al., 2011), aerosol
number (Leaitch et al., 2013, Croft et al., 2016a, b), aerosol absorption (Breider et al., 2014), mercury
(Fisher et al. 2012), and recently surface-layer DMS(g) (Mungall et al. 2016).
## 2.2.2  FLEXPART-ECMWF
For this study, the Lagrangian particle distribution model, FLEXPART model (Stohl et al., 2005; website:
https://www.flexpart.eu/) is driven by global meteorological analysis data from European center for
medium-range weather forecasts (ECMWF) for July 2014 and April 2015. For the ECMWF data a
horizontal grid spacing of 0.25° was used along with 137 hybrid sigma-pressure levels in the vertical from





1 the surface up to 0.01 hPa. FLEXPART was operated in backward mode to estimate potential emission

2 sources and transport pathways influencing Polar 6 DMS(g) measurements in summer 2014 and spring

3 2015.

## 3  Measurement results and discussion

### 3.1  Meteorological CO, aerosol number concentration and $O_3$ profiles

Wind interaction at the surface of the ocean, temperature, pressure, and ice cover are important factors in
DMS(g) exchange between the ocean and atmosphere. After emission from the ocean, the vertical profile
of DMS(g) is controlled by mixing in the boundary layer, air mass transport and chemical conversion
rates. Water vapour, $O_3$ and CO are potential indicators of DMS(g) conversion as the oceans are a source
of $H_2O$(g) as well as DMS(g). $H_2O$(g) and $O_3$ control OH production in the presence of sunlight, and CO
is an indicator of combustion and can be associated with other pollutant gas and aerosol transport. It is
interesting to examine the average particle concentration number as well since aerosol production under
clean Arctic conditions could be triggered when DMS(g) and sufficient oxidants are present. Alternately,
if the aerosols present were associated more strongly with CO and transport then DMS(g) may be reduced
due to oxidation on or within the surface of aerosols.
DMS(g) mixing ratios were compared with temperature, pressure, $H_2O$(g) and CO for each campaign and
show no evidence of any significant relationships (Figures S1 and S2). Figure S3a shows the vertical
profile for average $H_2O$(g) mixing ratios. Average $H_2O$(g) mixing ratios in the atmosphere were higher
during July 2014 (> 7300 ppmv) than April 2015 (> 900 ppmv), due to higher temperature, less ice cover,
and therefore more exchange between ocean and atmosphere.
However, when concurrent measurements of $H_2O$(g) during the flights is plotted against DMS(g), an anti-
correlation ($R^2$=0.6) is observed by considering both spring and summer campaigns together (Figure S4).
Assuming that both DMS and $H_2O$(g) originated from the ocean, which is much warmer than the Arctic
atmosphere in spring, loss of $H_2O$(g) relative to DMS(g) is implied by higher present of clouds in the
atmosphere during April. Cloud and precipitation would remove $H_2O$(g), but DMS(g) interacts weakly
with cloud water (Henry's Law constant of 0.14 mol/L-atm). Thus, more cloud processing of the





measured DMS(g) during spring, compared with summer, is implied. More cloud during transport may
also extend the DMS(g) lifetime by lowering the rate of photochemical oxidation.

## 3.2   DMS measurements and discussion

DMS(g) concentrations as a function of altitude are shown in Figure 4 for the July 2014 and April 2015
flights. The campaign-mean DMS(g) mixing ratios were 20±6  pptv (maximum of 114 pptv) for July 2014
and 116±8  pptv (maximum of 157 pptv) for April 2015.
The 2014 sampling locations focused on the Lancaster Sound, Nunavut region in July 2014, whereas
sampling in April 2015 occurred over a broad region of the Arctic: Longyearbyen, Spitzbergen, Alert and
Eureka, Nunavut and Inuvik, Northwest Territories. Observations on individual flights in July 2014
indicate either decreasing DMS(g) mixing ratios with increasing altitude or relatively uniform DMS(g)
mixing ratios (independent of altitude). During spring of the following year (April 2015),
DMS(g) mixing ratios on individual flights were more uniform with altitude and in some cases increased
with altitude.
During July, 2014, the highest DMS(g) mixing ratios were measured near ice-edges and above open
waters (e.g. samples >40 pptv, July 12, 20 and 21). That, and the decrease of atmospheric DMS(g) with
altitude, suggest the atmospheric DMS(g) was locally sourced (Lancaster Sound and Baffin Bay) during
the month of July, consistent with the findings of Mungall et al. (2016) conducted from the icebreaker
CCGS Amundsen. The decline in DMS(g) mixing ratios with height may be due to a combination of weak
vertical mixing and photochemical reactions.
Previous observations of seasonal variations in DMS(aq) in the Arctic Ocean found the maximum
DMS(aq) occurred in July and August (e. g. Bates et al., 1987; Leck and Persson, 1996, Levasseur 2013).
After the August peak, DMS(aq) declines due to lower biological activity (Leck and Persson, 1996). From
DMS(g) concentrations in both the surface ocean and in the atmosphere just above the ocean surface
(median DMS (g) of 186 pptv), Mungall et al. (2016) estimated the air-sea DMS(g) flux ranging from
0.02–12 μmol m$^{-2}$ d$^{-1}$ in July 2014 in the same location as the present measurements (Lancaster Sound).
For the same campaign, Ghahremaninezhad et al. (2016) showed that the dominant source for fine aerosol





and $SO_2$ measured onboard the Amundsen at the same location and about 30 m above the ocean's surface
was biogenic sulfur, arising from DMS(g) oxidation. Atmospheric oxidation of DMS(g) is expected to
proceed more readily in the summertime Arctic atmosphere than in spring, due to higher temperatures
and more sunlight.
During April, the higher DMS(g) mixing ratios, in the free troposphere over ice-covered regions (Figure
4, right panel), stability of the Arctic atmosphere, and limited vertical mixing, suggest that DMS(g) can
be transported to the sampling locations from other regions within the Arctic and/or from lower latitudes
(except for April 4 when DMS(g) sampling was above open water). These results contrast with results
from Ferek et al., (1995) where lower DMS(g) mixing ratios (a few pptv) were found over the Arctic
Ocean near Barrow during spring 1992. Andrea et al. (1988) presented vertical profiles of DMS(g) mixing
ratios measured over the northeast Pacific Ocean during May 1985 (with a maximum ~ 30 pptv in the
mixed layer and also 3600 m). They found that DMS(g) mixing ratios depend on the stability of the
atmosphere and air mass sources and that long-range transport at mid tropospheric levels was important
in remote areas of the northern hemisphere.
Hoffmann et al., (2016) showed that DMS(g) chemistry should be considered in both gas and aqueous
phases to improve modelling predictions. However, high DMS(g) mixing ratios aloft in spring with low
$H_2O(g)$ do not support strong DMS(g) oxidation in the aqueous-phase. Instead, the results shown here
are consistent with longer DMS(g) lifetimes in April than July due to lower OH mixing ratios enabling
long-range range transport of more DMS(g) (Li et al., 1993). Ozone depletion during spring was observed
within the boundary layer (Fig S3b) and is well documented in the literature (e.g. Barrie et al., 1989).
Ozone depletion may further decrease OH near the surface and enhance DMS(g) lifetimes in the boundary
layer. However, if present in the ozone-depleted boundary layer, halogen oxides, such as BrO radical,
are likely more important during winter/spring than summer and could oxidize DMS(g) (von Glasow et
al., 2004, Chen et al., 2016).
Average vertical profiles of aerosol number concentrations larger than about 5 nm diameter are shown in
Figure 5. The increase in number concentrations near the surface during July 2014 is coincident with the
higher levels of DMS(g) near the surface and highlights the role of the ocean (local source) in new particle
formation (NPF) during this study (Willis et al., 2016; Burkart et al., 2016). In contrast, elevated particle





number concentrations during April 2015, were more often seen aloft with a decrease towards the surface.
With the current dataset, it is impossible to say if potential NPF events aloft during the spring were
connected with the higher DMS(g) aloft.

## 5  4   Chemical-transport-model simulations and discussion

## 6  4.1   GEOS-Chem

We simulated the vertical profile of DMS(g) mixing ratios with the GEOS-Chem chemical-transport
model, and the model was co-sampled along the Polar 6 aircraft tracks. Recently, GEOS-Chem was used
to interpret DMS(g) measurements in the Arctic surface-layer atmosphere (Mungall et al., 2016).
However, despite the significant influence of DMS(g) on the Arctic climate relative to lower latitudes and
the importance of where DMS(g) oxidation occurs vertically (Woodhouse et al., 2013), measurements of
DMS(g) vertical profiles are rare in the Arctic atmosphere.
Figure 6 shows the campaign-mean vertical profile of DMS(g) for the co-sampled GEOS-Chem
simulation and our measurements for both July 2014 and April 2015. In July 2014, both the measurements
and simulation show a strong decrease of DMS(g) mixing ratios with altitude in the lowest 300 m.
Aliabadi et al. (2016a and 2016b) estimated the boundary layer height as 275±164 m, using data from
radiosondes launched at Resolute Bay and the Amundsen icebreaker, during the 2014 campaign. Aliabadi
et al. (2016b) indicated that the magnitude of turbulent fluxes of momentum, heat and the associated
diffusion coefficients are significantly reduced above the boundary layer height during the 2014
campaign. Thus, we find the strongest vertical gradient between the boundary layer and above. In the
boundary layer, the GEOS-Chem simulation over predicts the measurements, but is within a factor of 2
to 3. Above 1500 m, the simulation under predicts the measurements.  Overall, the simulations and
observations agree within their respective uncertainties.
The April 2015 campaign-mean shows a more gradual decrease with altitude (Figure 6, right panel). Both
the simulated and measured DMS(g) profiles show less variability with altitude than in summer. As well,
mixing ratios are greater than during the July campaign. Ozone depletion not represented by the
simulation is one potential explanation for the underestimated DMS(g) in the simulations. Surface layers





depleted of ozone were observed on several occasions during April 2015: 3 of 5 samples collected at 60
m above ice surfaces were concurrent with measured ozone depletion events (<1 ppbv) during the April
campaign. If the DMS(g) oxidation potential is reduced by ozone depletion the lifetime of DMS(g) in the
region of ozone depletion may increase. Another reason for underestimation by the model may be errors
in the source strength.
We conducted a sensitivity simulation to identify the latitude-dependent contribution of the oceans to the
simulated DMS(g) at the sampling points along the flight tracks. In Figure 6, the "SimZeroBelow66"
simulation has no ocean DMS(g) for all latitudes south of 66°N. This simulation compared with the
standard simulation suggests that a large majority of the campaign-mean DMS(g) for both April and July
arises from the oceans north of 66°N.
As given in Table 2, the 'SimZeroBelow66' simulates 97% or more of the DMS(g) below 500 m during
July coming from waters north of 66°N. The fractional contribution from north of 66° is about 90% for
April and at the same altitudes; although different regions were sampled at that time. The simulations
attribute about 60% and 90% of the DMS(g) at altitudes from 500-3000 m to seawater north of 66°N in
April and July, respectively. This 30% difference indicates a greater contribution from long-range
transport from lower latitudes in the springtime.

## 4.2  FLEXPART

FLEXPART-ECMWF modeling was used to explore the origin of air mass back trajectories for the Polar
6 flight tracks.
Figure 7 shows two examples of FLEXPART-ECMWF potential emissions sensitivity (PES) for 4-day
back trajectories on July 2014: an influence from a broad area and especially Lancaster Sound (local
region) and north on July 12th (Figure 7(a)), and Hudson Bay, and Baffin Bay (south) on July 19th (Figure
7(b)). The PES analysis shows the air mass descended from >1500 m on July 19th, which may explain the
lower DMS(g) mixing ratios.
Figure 8 shows some examples of FLEXPART-ECMWF PES simulations for 4-day back trajectories
during April 2015. For the flights near Alert and Eureka on April 9 and 11, the potential emissions





originated from the northwest of Greenland (Figure 8(a) and Figure 8(b)). For the April 13 flight, the
Norwegian Sea and the North Atlantic Ocean are additional potential source regions (Figure 8(c)), and
for the April 20 flight near Inuvik, with the highest measured DMS mixing ratio, the potential emissions
originated from North Pacific Ocean (Figure 8(d)). The April 20th flight was the only flight south of
~80°N during the springtime and showed a, large influence from mid-latitudes.
In general, these results suggest that the DMS(g) measured during July 2014 originated primarily from
the local region over Baffin Bay and the Canadian Arctic Archipelago. For spring 2015, the DMS(g)
sampled was from a range of sources, including Baffin Bay, possibly the Norwegian Sea, the North
Atlantic Ocean and the North Pacific Ocean.

## 5   Conclusion

Atmospheric samples for DMS(g) measurements were collected at different altitudes aboard the Polar 6
aircraft expeditions during July 2014 and April 2015, as part of the NETCARE project. In this study, we
present vertical profile measurements of DMS(g), together with model simulations to consider what these
profiles indicate about Arctic DMS(g) sources and lifetimes. Vertical variations in DMS(g) mixing ratios
will likely influence aerosol concentrations via new particle formation and growth, which could impact
Earth's radiation budget.
Our measured vertical profiles of DMS(g) suggest differences between the main sources and lifetime of
DMS(g) during the Arctic summer and spring. For the summertime flights near Lancaster Sound,
Nunavut, Canada, DMS(g) mixing ratios were higher near the surface (maximum > 110 pptv) and lower
at higher altitudes up to 3 km. The highest mixing ratios were found above ice edges and open waters
suggesting that the Arctic Ocean in the vicinity of the aircraft was the main source of DMS(g). Oxidation
and/or limited vertical mixing could contribute to the decline of DMS(g) mixing ratios with altitude.
During the springtime pan-Arctic flights from Svalbard to the Canadian Arctic Archipelago and ending
near Inuvik, Northwest Territories, the measured DMS(g) mixing ratios were unusually high (> 100 pptv),
and more uniform with altitude than during summer. DMS(g) mixing ratios in samples collected in the
free troposphere (>2000 m) during April ranged from 60-134 pptv. Transport of DMS(g) to the high-



Arctic from other regions of the Arctic and/or at lower latitudes with reduced oxidizing potential may
explain these observations.
The DMS(g) vertical profile along the flight tracks was simulated with the GEOS-Chem chemical
transport model. The measurement and simulated co-sampled campaign-mean DMS(g) vertical profile
agreed within a factor of 3 for both July 2014 and April 2015. A sensitivity test indicated that the oceans
north of 66°N contributed about 97% and 90% of simulated DMS(g) at altitudes below 500 m at the
measurement sampling times in July and April, respectively. For the April flights, about 60% of the
simulated DMS at altitudes between 500-3000 m was attributed to water north of 66°N. Potential emission
sensitivity from FLEXPART analysis for the aircraft tracks showed that local sources (Lancaster Sound
and Baffin Bay) primarily contributed to air sampled during July 2014. On the other hand, long-range
transport (LRT) from the northern tip of Greenland of air that originated over the waters to the northwest
of Greenland as well as the North Pacific Ocean were important contributors to air masses sampled during
April 2015.
In short, this study suggests a dominant role of the Arctic Ocean for DMS(g) in the Arctic during summer,
and a significant contribution from LRT to DMS(g) in spring.
**Acknowledgment**
This study was part of the NETCARE (Network on Climate and Aerosols: Addressing Key
Uncertainties in Remote Canadian Environments, http://www.netcare-project.ca/) and was supported by
funding from NSERC. The authors also would like to thank the crew of the Polar 6 and fellow
scientists. Data is available by email request (alnorman@ucalgary.ca). The GEOS-Chem model is freely
available for download from www.geos-chem.org.



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





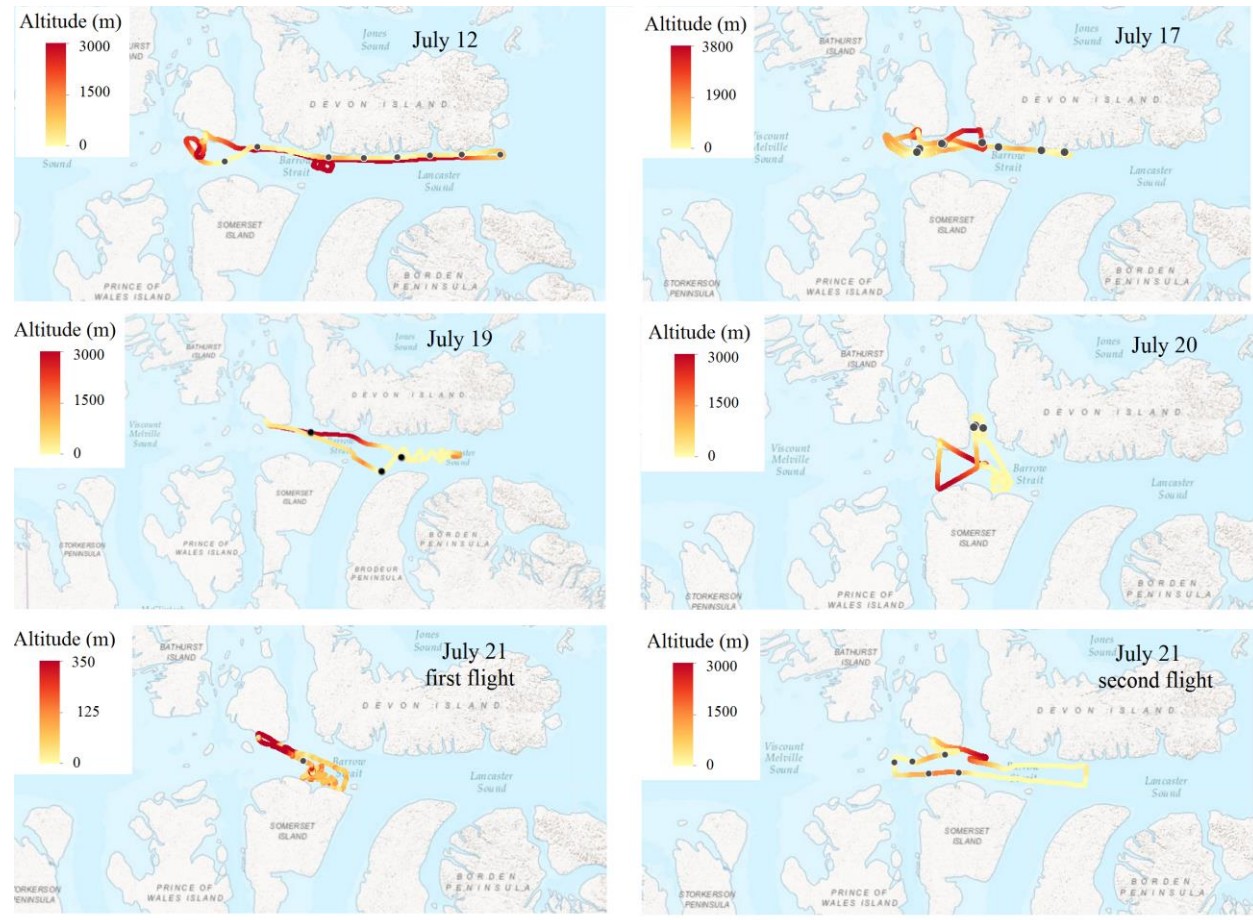

Figure 1. Polar 6 aircraft routes from 12 to 21 July 2014. Color bars indicate altitudes and sampling locations are shown with black dots.

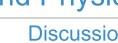

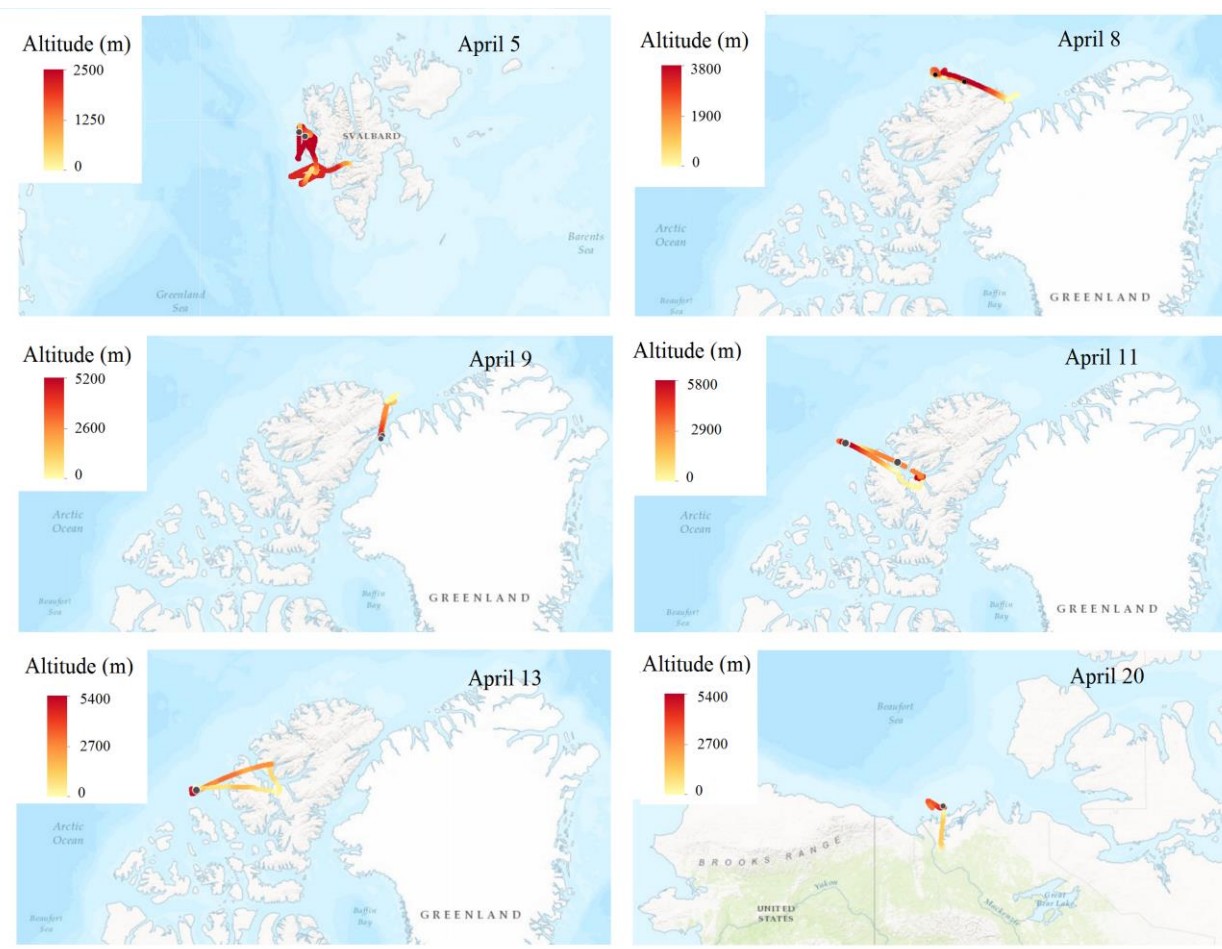

Figure 2. Polar 6 aircraft routes from 5 to 20 April 2015. Color bars indicate altitudes and sampling locations are
shown with black dots.



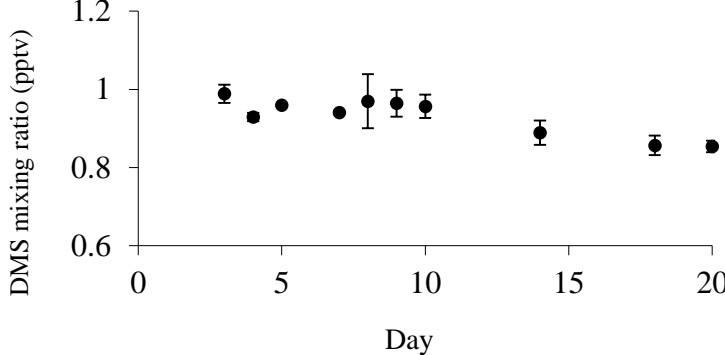

Figure 3. DMS mixing ratios versus Tenax storage days. Error bars indicate the standard deviation for each test.






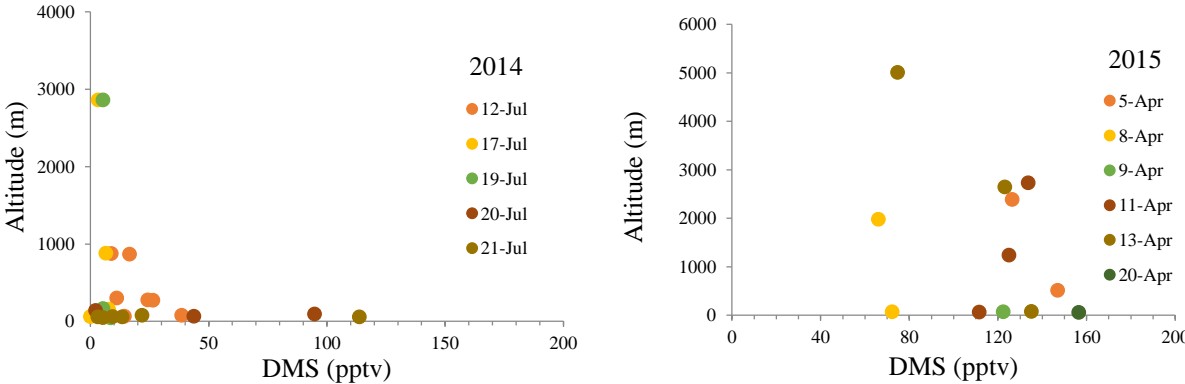

Figure 4. Sampling altitudes (m) versus DMS mixing ratios (pptv) for July 2014 (left panel) and April 2015 (right panel).





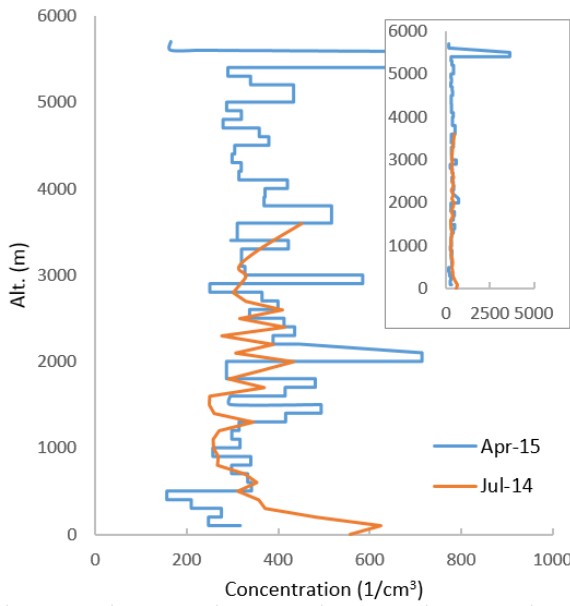

Figure 5. Average vertical profile of particle number concentration for July 2014 (orange), and April 2015 (blue). The insert shows a peak in concentration at high altitude (>5000 m) for April 2016.





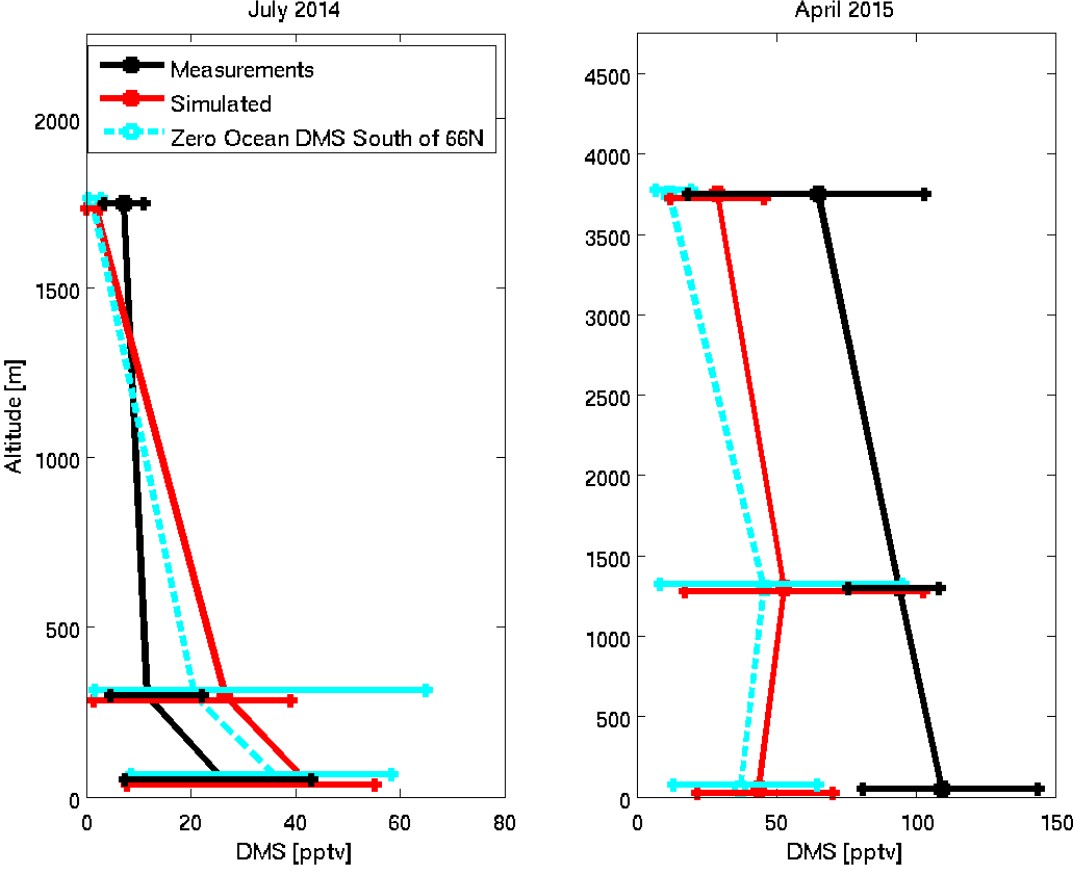

Figure 6. The campaign-mean vertical profile of DMS from the GEOS-Chem simulation (red line) and measurements (black line) for July 2014 and April 2015. Simulations for zero ocean DMS at latitudes south of 66°N (SimZeroBelow66) are shown as cyan dashed line. The 20[th] and 80[th] percentiles are shown by horizontal bars.




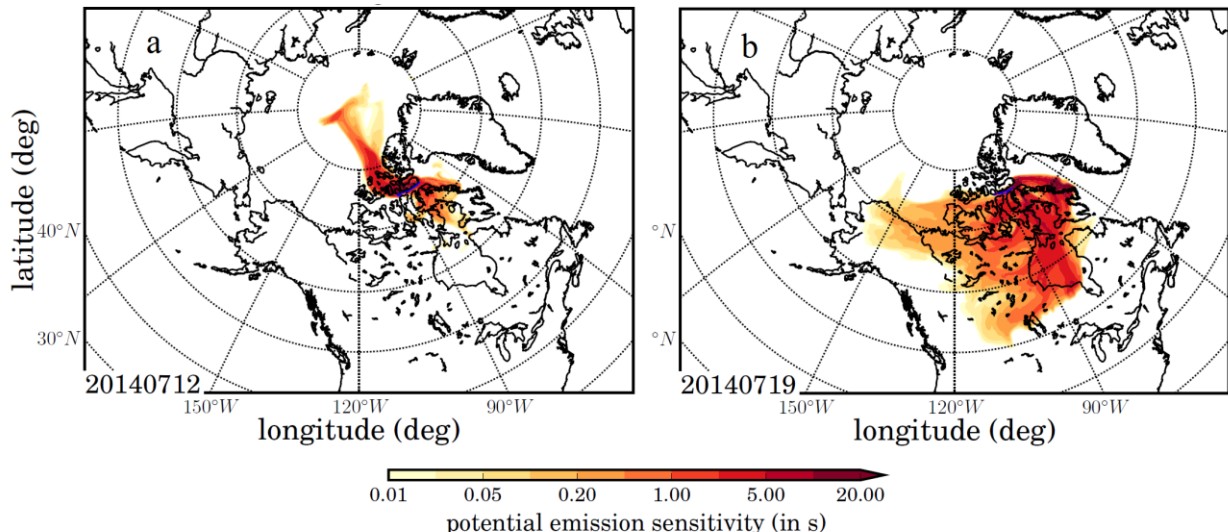

Figure 7. FLEXPART-ECMWF potential emissions sensitivity simulation plots for 4-day back trajectories for column from 0 to 200 m on (a) July 12$^{th}$ (20:40:00 h UTC) and (b) July 19$^{th}$ (17:00:00 h UTC), 2014. The color bars indicate air mass residence time (seconds) before arriving at the aircraft location. The blue lines show Polar 6 aircraft routes.





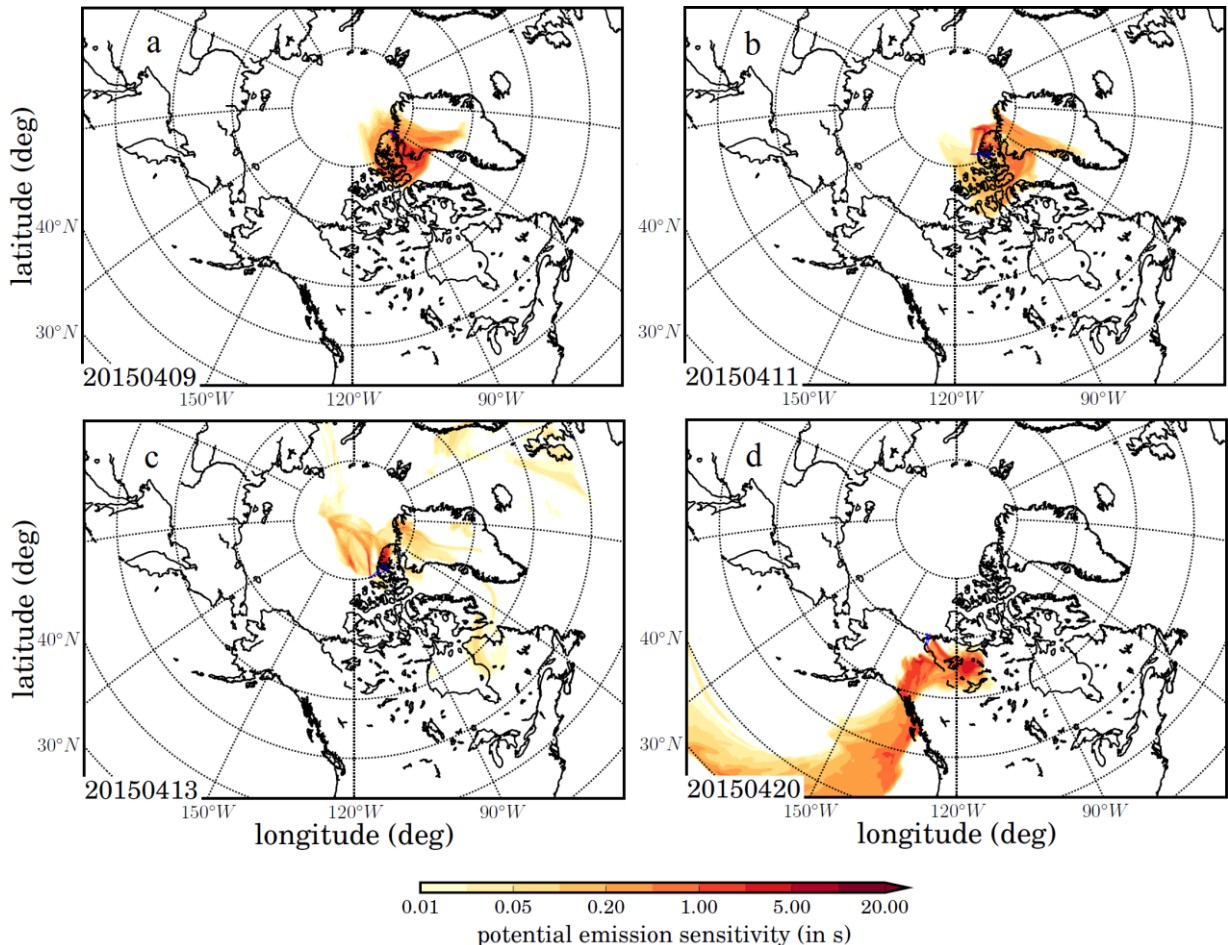

Figure 8. FLEXPART-ECMWF potential emissions sensitivity simulation plots for 4-day back trajectories for column from 0 to 200 m on (a) April 9th (14:45:00 h UTC), (b) April 11th (18:55:00 h UTC), (c) April 13th (18:27:00 h UTC), and (d) April 20th (22:26:00 h UTC), 2015. The color bars indicate air mass residence time (seconds) before arriving at the aircraft location. The blue lines show Polar 6 aircraft routes





Table 1. DMS mixing ratio values, sampling and analysis dates for July 2014 and April 2015.

| Sample # | DMS (ppt) | Sampling Day | Analysis Day | Sample # | DMS (ppt) | Sampling Day | Analysis Day |
|---|---|---|---|---|---|---|---|
| 1 | 17 | 12/07/2014 | 25/07/2014 | 20 | 2 | 20/07/2014 | 25/07/2014 |
| 2 | 39 | 12/07/2014 | 25/07/2014 | 21 | 22 | 21/07/2014 | 25/07/2014 |
| 3 | 26 | 12/07/2014 | 25/07/2014 | 22 | 5 | 21/07/2014 | 25/07/2014 |
| 4 | 14 | 12/07/2014 | 25/07/2014 | 23 | 3 | 21/07/2014 | 25/07/2014 |
| 5 | 24 | 12/07/2014 | 25/07/2014 | 24 | 13 | 21/07/2014 | 25/07/2014 |
| 6 | 9 | 12/07/2014 | 25/07/2014 | 25 | 114 | 21/07/2014 | 25/07/2014 |
| 7 | 11 | 12/07/2014 | 25/07/2014 | 26 | 9 | 21/07/2014 | 25/07/2014 |
| 8 | 6 | 12/07/2014 | 25/07/2014 | 27 | 127 | 05/04/2015 | 07/05/2015 |
| 9 | 5 | 17/07/2014 | 27/07/2014 | 28 | 147 | 05/04/2015 | 07/05/2015 |
| 10 | 8 | 17/07/2014 | 27/07/2014 | 29 | 66 | 08/04/2015 | 07/05/2015 |
| 11 | 8 | 17/07/2014 | 27/07/2014 | 30 | 72 | 08/04/2015 | 07/05/2015 |
| 12 | 6 | 17/07/2014 | 27/07/2014 | 31 | 122 | 09/04/2015 | 07/05/2015 |
| 13 | 6 | 17/07/2014 | 27/07/2014 | 32 | 134 | 11/04/2015 | 08/05/2015 |
| 14 | 3 | 17/07/2014 | 27/07/2014 | 33 | 112 | 11/04/2015 | 08/05/2015 |
| 15 | 5 | 19/07/2014 | 27/07/2014 | 34 | 125 | 11/04/2015 | 08/05/2015 |
| 16 | 5 | 19/07/2014 | 27/07/2014 | 35 | 75 | 13/04/2015 | 08/05/2015 |
| 17 | 9 | 19/07/2014 | 27/07/2014 | 36 | 123 | 13/04/2015 | 08/05/2015 |
| 18 | 44 | 20/07/2014 | 25/07/2014 | 37 | 135 | 13/04/2015 | 08/05/2015 |
| 19 | 95 | 20/07/2014 | 25/07/2014 | 38 | 157 | 20/04/2015 | 08/05/2015 |






Table 2. Simulated campaign-mean percent contribution of DMS from oceans north of 66°N to the GEOS-Chem
simulated DMS at the sampling locations for the July 2014 and April 2015 flight tracks.

| Altitude | July 2014 | April 2015 |
|---|---|---|
| 0-100 m | 98 | 88 |
| 100-500 m | 97 | 90 |
| 500-3000 m | 91 | 61 |
