# Peer review of "Vertical profile of atmospheric dimethyl sulfide in the Arctic"

_Atmospheric Chemistry and Physics, 2017_

## Referee Comment (RC1) · Anonymous Referee #1 · 10 Feb 2017

This paper addresses aspects of the linkage between surface DMS emissions and Arctic CCN/cloud dynamics. Arctic clouds play a critical role in the surface radiation balance and thus sea ice dynamics. The prevalence of stable atmospheric conditions over sea ice can inhibit a direct connection between local surface emissions and clouds, thus long range transport and downward-mixing of CCN and aerosol precursors may be more important, as in Lunden et al. (2010). See Shupe et al. (2013) for further evidence that long range transport of moisture, CCN, etc. are important for arctic cloud dynamics (doi:10.5194/acp-13-9379-2013).

Studies referenced by the authors reveal that seasonal DMS emissions in the marginal ice zone (MIZ) and open water can be quite large, with significant variability in space

and time. Diagnosing a link between these emissions and aerosol/cloud dynamics ultimately requires CTM simulations (and physically realistic, validated cloud/aerosol models!). This appears to be the authors intention here, but to me their approach seems backwards. They use modeling to explain the source of DMS measured on their flights. Most of the measurements are close to the surface and presumably within the atmospheric boundary layer (ABL), over ice covered seas, the MIZ and open water. Except for flights over ice covered seas in spring, the source of DMS near the surface is less interesting that the fate of that DMS, since the source is most certainly local but the potential effects on aerosol and cloud dynamics may be far afield.

I'm not sure section 3.1 provides much insight. It's not clear what 'cloud processing of DMS' means, and correlations of DMS with water vapor, CO and ozone are subject to such a wide variety of dynamic and chemical influences that they are difficult to interpret. Conclusions here are vague. Fig. S4 merely reveals that DMS emission peaks in spring and water vapor peaks in summer, which is expected based on existing information. The vertical distribution of all these species is much more interesting. The authors are using an advanced CTM to simulate vertical distribution and transport. I'd like to see model results for H2O, CO and O3 plotted with the observations on Fig. S3, in a manner similar to that shown for DMS in Fig.6. It's important to know how well the model reproduces the vertical distribution of these tracers. I think the authors should include a discussion of the vertical structure of the atmosphere in this section. Is there a well-defined atmospheric boundary layer on each flight? At what height? Does the profile of potential temperature, water vapor or vertical wind velocity indicate a well-mixed ABL and strong inversion? Is there evidence of atmospheric stability or of convection and mixing into the free troposphere? What does the CTM assume or simulate for vertical mixing on the flight dates and how does that compare with the observed profiles? How do profiles differ between open water, MIZ and ice covered regions? On p.10 the authors talk about limited vertical mixing and atmospheric stability during the April flights, but present no data or analysis to support this statement.

In all studies of atmospheric DMS over the ocean in the absence of deep convection, the concentration of DMS is elevated and well-mixed below the ABL inversion, dropping to near zero above the inversion. This is due to limited vertical transport and a relatively short photochemical lifetime. If NETCARE results differ from the usual DMS vertical distribution over the ocean, that's an interesting finding, implying convective transport into the free troposphere and perhaps an extended photochemical lifetime due to reduced water vapor or limited sunlight, all of which can be diagnosed with the CTM. In this regard, the vertical profile results for DMS in Section 3.2 are quite interesting. From Fig.4a it looks like atmospheric DMS is often elevated and highly variable near the surface in mid-summer. At 3000m the concentrations are near zero and I presume this represents the background free troposphere (are these measurements at the detection limit?). But, the few measurements on Jul12/17 at 1000m are significantly elevated. Are these within a deep ABL or are they evidence of long range transport in the FT? The latter would be very interesting.

Fig. 7 shows CTM back trajectories for the 0-200m level, so they presumably apply to aircraft measurements at the lowest altitude only. Are the colors in Figs. 7/8 meant to signify surface source regions for low-level air sampled by the aircraft? I don't understand what is meant my 'air mass residence time in seconds before arriving at the aircraft location'. The authors should define 'potential emission sensitivity'. Also, the blue line indicating the flight path is not visible and probably too small properly plot on a map of this scale. Maybe it's best to just put a dot on the map indicating the flight location. Are the authors suggesting DMS measured near the surface over the MIZ and open water in mid-summer near Devon Island originated as far away as Hudson Bay? That doesn't make sense to me. So I'm not sure about the significance of Figs. 7 and 8. It would be more interesting if these were forward trajectories rather than back trajectories.

For July flights over the ice edge and open water in mid-summer (a time of extended daylight hours and perhaps greater photochemical oxidation) the author's conclusion

that DMS sampled near the surface represents local emissions is quite reasonable. DMS within the ABL at 0-200m is likely to have originated from surface emissions quite near the sampling site under these conditions. I would be much more interested in back trajectories for measurements at the 1000m level, especially if this is above the ABL. If not, then the statement on p.9 that 'The decline in DMS mixing ratios with height may be due to a combination of weak vertical mixing and photochemical reactions' is entirely comparable to conditions that exist over the ocean in most other locations in the world. This seems to be supported by the CTM results in Fig. 6. I would think they could provide evidence of vertical structure in the atmosphere to further support this conclusion.

For April flights, the measurements indicate high DMS concentrations near the surface, which is understandable given algal blooms associated with the spring melt, but also elevated concentrations aloft, which is very interesting. It would be good to know the fractional ice coverage for the April flights and if the surface exhibited extensive melting with open leads. We also need to see an analysis of the vertical structure of the atmosphere for these flights. Fig. 8 should show back trajectories for the upper flight levels to help diagnose the significance of long range transport and potential source regions. This seems more relevant than the source of surface atmospheric DMS, which is likely to be local and may be inhibited by atmospheric stability from contributing CCN and cloud dynamics aloft. Though it would be interesting to see a forward model trajectory for the surface air and an analysis of the potential to influence to aerosol aloft over remote, ice covered regions. Heat fluxes in the MIZ, where open water is present, might drive sufficient convection to transport surface emissions from the MIZ and open water to ice-covered regions, where cloud formation is driven by mixing from above. Hopefully, the CTM can reproduce these processes.

In summary, although the amount of data is limited, I think the authors have interesting measurements from an important region of the world, and they have a very powerful modeling system at their disposal. But I don't think they have done sufficient analysis

of the specific meteorological and atmospheric conditions on each flight for me to fully understand the observations, and I don't think they've made best use of the model's capabilities to diagnose the fate of DMS in the Arctic ABL. I'm very interested in this topic and look forward to a revised version of this contribution.

---

## Referee Comment (RC2) · Anonymous Referee #2 · 26 Feb 2017

This paper discusses seasonal measurements of DMS in the Arctic atmosphere with special emphasis on vertical distributions and evaluation of sources and impact. I have to say, though, given the title, that I was surprised to see only a handful of the measurements (between 10 – 12 samples total in 2 separate campaigns) were obtained at altitudes greater than 500 m. Given the variability and uncertainty in DMS emission intensity and distribution, this seems to be a marginal data set for evaluation by the modeling tools that were applied here. Furthermore, the measurements are obtained in a very limited geographic range of the Arctic and may not be particularly representative of the Arctic. Still, given the paucity of reported airborne measurements of DMS in the Arctic, there may be interest to see these data presented and discussed, otherwise they just disappear into some data repository.

First, I think the description of the measurement technique needs some clarification, even though it is based on methods described 20 years ago. My main questions to be cleared up: 1) Sample times range from 2 – 11 minutes. What is the flow rate, and what are the resulting sample volumes? 2) Reference to Sharma et al. (1999) as a description of the method seems incorrect as listed. The listed Sharma et al. (1999) paper describes a 10 L sample collected on molecular sieve. The reference Sharma et al. (1997) is a typo and should also be 1999. This appears to be the correct reference to the Tenax method. Sharma (1997) is the MSc thesis, which should contain all of the details but is not easily available which is why some extra detail is needed. The Sharma et al. (1999) paper lists the detection limit as -+6 pptv in a 2 L sample (defined as 2:1 S/N), and accuracy and precision of -+12% (not sure what this means, though). This manuscript refers to a -+ 12 pptv uncertainty (not specified if accuracy or precision). Surely this can't be constant for samples of different volumes. Was this determined based on the triplicate standard measurements, or was this somehow referenced back to Sharma et al. (1999)? Please clarify this, esp. since many of the samples reported seem to be in this range. 4) Sample storage tests: how much standard was loaded onto the cartridge? Were different loadings tested? Please also note the axis of the graph in Figure 3. Is this really the DMS mixing ratio in pptv, or some ratio to initial DMS addition? Is the uncertainty in the stability also considered in assessing the overall uncertainty of -+12 pptv?

P7, L.22. The GEOS-CHEM model is certainly widely used for many applications, but as I looked at the Mungall et al., 2016 paper, it seemed that there were significant issues with the DMS emissions specific for the region that was studied, and these conclusions suggest that the GEOS CHEM model needs to be applied with caution for specific locations and seasons for dealing with a highly variable compound such as DMS. Just because it was used is not necessarily an endorsement for its applicability for these measurements, at least without some caveats.

P8, L6. Meteorology and profiles. It is unusual to see the vertical profiles of species

averaged for an entire campaign. The DMS measurements need to be related to the atmospheric vertical structure and variability during each flight, or there should be some demonstration that a more suitable analysis is found by taking mission averages.

P8, L 16. Please provide the reference(s) to the heterogeneous oxidation of DMS on or within aerosols.

P8, L 24. While sources of DMS and H2O are the ocean (though there could be other sources of DMS), different correlations could be observed due to the high spatial and temporal variability of DMS (compared to H2O) and effect of temperature on either gas. I assume this discussion of the correlation is related to the argument about the greater presence of clouds in spring vs. summer, and (again) the assertion of a major role of cloud processing of DMS. The presence (note typo on P8, L25) of clouds could potentially have a greater impact on OH production below or above the clouds with subsequent impact on DMS lifetime. Perhaps this was suggested by the statement regarding clouds during transport, but this seems a speculation that could be checked from observations. It is not clear what the authors are saying regarding the effect of clouds....reduction of DMS due to heterogeneous O3 oxidation, or increase of DMS due to slower OH oxidation? Unfortunately, the data is not available to support either impact so this just remains as speculation.

P9, L 25. The authors reference again the work reported by Mungall et al., 2016 who measured DMS at the same general location and time period (as far as I can tell) as the airborne data for July, 2014. It would be helpful to discuss the airborne measurements and the impact of the very large variability in DMS emissions that were found in Mungall et al. Perhaps there were times of reasonable overlap between the ship based and airborne measurements that could also be compared. It seems that the large DMS mixing ratios observed during parts of the Mungall et al. cruise were not sampled by the aircraft. Furthermore, the reported range of a factor of 60 in the calculated DMS flux should further raise a cautionary flag about how well climatological models and emission estimates might be applied to the data.

P10, L 19. Here and elsewhere in the manuscript the possibility of long-range transport of DMS is mentioned. In this context, it would be helpful if the authors could provide some reasonable estimate of the lifetime of DMS and the possible transport times that are considered "long-range" vs. local.

P10, L 25. I am not a fan of averaged vertical profiles for variable species (such as aerosols) without some indication of the variability of the measurements over the different profiles. I assume that the higher particle number is associated with the specific profile in July where the measured DMS is about 100 ppt. Or is the higher level of aerosol more widespread? Also, the springtime levels of DMS are even higher than those found in summer, but no significant aerosol production here? This level of discussion and interpretation is incomplete and not particularly informative.

P11, L 3. Thin aerosol layers are common in the Arctic spring. It seems very unlikely that the high altitude peak in aerosol is in any way related to DMS. If the authors really consider the DMS source likely, perhaps some analysis of what sort of conditions might be necessary to produce the observed aerosol abundance.

P11, L 13. Do I understand correctly that the GEOS-CHEM results shown in Figure 6 are compared only for the 2 – 10 minute sample integration time of the measurements? It would be interesting to see some examples of the variability of GEOS CHEM results to perhaps provide a more detailed context for the very limited number of actual samples that were collected. Often models track measurement features, but may be slightly off in time or altitude.

P11, L 23. I am not sure what the authors want to convey when they claim that the GEOS CHEM data and the measurements are "within their respective uncertainties". Perhaps "variability" is a better term than uncertainties in this case. There are clearly more uncertainties in the model results beyond just the calculated mixing ratios.

P11, L25. From Figure 6, the variability with altitude actually seems less in the summer compared to spring, though the gradient is less in the spring, as noted. Summer range

about 20 – 40 ppt at low altitudes, <10 ppt at max altitude; Spring 30 – >50 ppt at all altitudes. Could the authors please quantify the difference in seasonal variability at different altitudes to demonstrate their assertion?

P12, L16. 30% difference might also be due to incorrect distribution and intensity of model DMS emissions.

P13, L5. I am concerned about the conclusions from the Flexpart analysis where the authors claim a large influence for specific regions. As noted, these analyses suggest where the sampled air spent time in the boundary layer. This analysis must be combined with an emission distribution to provide the source attribution. For example, without additional information, it is misleading to suggest that North Pacific was a significant source of DMS measured in the Arctic. It is a potential source. If the authors have some other data (ocean color, for example) to indicate DMS emission in the North Pacific at that time, then that could be a basis for the claim. Without this emission information, the Flexpart analysis can only indicate "potential" regions.

P13, L9. Could the authors make some estimate, based on typical DMS lifetimes, what the source region DMS mixing ratios would need to be to support the observed mixing ratios in spring?

P13, L16. The link established in this data set between aerosol formation and DMS is very weak and essentially presented as a given with no detailed analysis. While we all seek global significance of atmospheric chemistry and a potential link to climate, I think that it is an overreach to state that, without much greater data coverage and analysis, the observed vertical variation will significantly impact the Earth's radiation budget.

Figures 1 and 2. These are not particularly helpful figures to identify the sample locations. I would suggest combining each season's flights into several composites that showed 1) a map of flight tracks, 2) an altitude-longitude (and/or latitude) cross-section that identified the location of the samples and the flight tracks. Given only a few flights, a color code could make things relatively easy to follow.

Figure 3. Already noted..Vertical axis mislabeled?

Figure 5. Possible to add shading to each mean profile to show range or std deviation in each bin?

Figure 7. I could not distinguish the blue lines for the flight tracks. Are these calculated for a single sample or some altitude? Please specify. I certainly do not understand the units for potential emission sensitivities. I thought that these indicated the time spent in the boundary layer (in this case 0 – 200 m) during the period of the back trajectories (4 days here). Is there a factor of 1000 missing here?

Table 1. It would be helpful to add the flight number and altitude to this table.

―――――――――――――――――――

---

## Referee Comment (RC3) · Anonymous Referee #3 · 9 Mar 2017

This is an interesting study of DMS profiles obtained at different times of the year and at different locations. The introduction is a nicely written summary of Arctic DMS emissions, chemistry, and potential impacts on climate, etc.

The overall methodology is probably good but the description of the DMS measurements is confusing and would benefit from rewriting and reordering some sentences. Why did the sample collection times vary so much? Is there a reason for this? It seems like different collection times will result in different amounts of sample collected resulting in different limits of detection. Please comment on this and clarify. The paragraph beginning on page 5 line 23 is particularly confusing. It seems that this paragraph was intended to describe the calibration methodology but this is not obvious. It is stated

that "Three Tenax tubes were injected with standard DMS along with one blank Tenax tube for each test period...", Why? What is the meaning of this? This is followed by a statement about calibrating the GC-SCD with 1 and 50 ppmv gas DMS standards. Where did the laboratory get these standards. Were they certified standards etc. Collection and analysis were referenced to Sharma and Rempillo after the collection was already briefly described prior to this statement. It is stated that the uncertainty is $\pm 12\%$ with this method but is that somehow independent of the amount of sample collected AND the mixing ratio of the sample that was collected? Please clarify and add a brief description of the Sharma and Rempillo methods and how the 12% uncertainty is determined.

The Tenax storage test shown in Figure 3 needs further discussion. The authors prepared a 1 pptv sample which is impressive. Would like details on how they did that. It is not clear what the standard deviation for each test represents. How many times were the samples analyzed etc.? And does the test have meaning given the uncertainty in the measurements? What is the LOD of the measurements?

DMS measurements and discussion – the decline in DMS mixing ratios with height in July is essentially what is expected and the pattern been seen in a number of previous studies. As stated, it results from primarily from fast photochemical destruction in the absence of deep convection as the lifetime of DMS is fairly short in July ($\tau \approx 1$ day). The data points above the surface (1 and 3 km) could be interesting but it would be instructive to know/understand the confidence that the authors have in these measurements with respect to LODs etc. Also related to that, I agree with other reviewers that the authors should include more discussion of the vertical structure of the atmosphere in section 3.1 It is important to know if there evidence of atmospheric stability or of convection and mixing into the free troposphere.

The April results are definitely quite interesting. I am surprised at both the surface measurement and aloft mixing ratios since this time of year is early for substantial biological productivity I would think. The authors didn't mention previous observations from the NASA DC-8 during ARCTAS (https://www-air.larc.nasa.gov/cgi-bin/ArcView/arctas). The results in paper contrast with the ARCTAS data in spring where lower DMS mixing ratios were observed (below detection limit to a few pptv and a max of 1 pptv in the free troposphere). It would be interesting to describe leads observed etc. in the region of sampling.

I am curious about the DMS emission source inventories used in the model and where these came from during springtime.

In summary, the paper presents some interesting data. I am a little concerned about drawing too firm conclusions from a relatively sparse data set. But if the authors can give further evidence of the robustness of their technique including limits of detection and a better description of blanks and uncertainties etc., providing a firm defense of their data, then the conclusions and/or hypotheses given in the paper can be given more weight and in that case the results should be published because the observations, if they hold, up are of interest to the community and relevant to atmospheric chemistry with possible climate implications.

Minor things:

P3 line 8 – describe CLAW hypothesis

P4 line 13 add s to altitudes to make it plural

P4 line 20 – suggest replacing "act" with "appear"

P5: As suggested above rewrite paragraph beginning on line 17

P8 line 25 replace "higher present" with "a higher presence"

P8 line 26 – make Cloud plural – "Clouds"

P10 line 5 eliminate comma after mixing ratios

---

## Author Comment (AC1) · 25 Apr 2017

The comment was uploaded in the form of a supplement:
https://acp.copernicus.org/preprints/acp-2017-33/acp-2017-33-AC1-supplement.zip

---

## Author Comment (AC2) · 25 Apr 2017

The comment was uploaded in the form of a supplement:
https://acp.copernicus.org/preprints/acp-2017-33/acp-2017-33-AC2-supplement.zip

---

## Author Comment (AC3) · 25 Apr 2017

The comment was uploaded in the form of a supplement:
https://acp.copernicus.org/preprints/acp-2017-33/acp-2017-33-AC3-supplement.zip

---

## Author Response (AR1)

Dear Prof. Russell,

We thank you and reviewers for your time and the great comments. We believe that we have answered all the comments/concerns.

Please find the supplement: Supplement includes comments, our responses (in italics), and the modified text in the manuscript (highlighted).

Your Sincerely,

Roghayeh Ghahremaninezhad (Roya)

PDF, Air quality Research Division

Environment and Climate Change of Canada

Toronto, Canada

Tel: 1-416-739-4690

E-mail: Roghayeh.Ghahremaninezhad@canada.ca

**Referee 1**

This paper addresses aspects of the linkage between surface DMS emissions and Arctic CCN/cloud dynamics. Arctic clouds play a critical role in the surface radiation balance and thus sea ice dynamics. The prevalence of stable atmospheric conditions over sea ice can inhibit a direct connection between local surface emissions and clouds, thus long range transport and downward-mixing of CCN and aerosol precursors may be more important, as in Lunden et al. (2010). See Shupe et al. (2013) for further evidence that long range transport of moisture, CCN, etc. are important for arctic cloud dynamics (doi:10.5194/acp-13-9379-2013).

*Thank you- We refereed to Shupe et al. (2013).*

(Page 2, line 23) Shupe et al., (2013) provided the evidence for the formation of clouds and transport of moisture and aerosol particles, likely accompanied warm air masses, from lower latitudes into the central Arctic during summer.

Studies referenced by the authors reveal that seasonal DMS emissions in the marginal ice zone (MIZ) and open water can be quite large, with significant variability in space and time. Diagnosing a link between these emissions and aerosol/cloud dynamics ultimately requires CTM simulations (and physically realistic, validated cloud/aerosol models!). This appears to be the authors intention here, but to me their approach seems backwards. They use modeling to explain the source of DMS measured on their flights.

*Please note that we tried to compare DMS(g) mixing ratio (and source) during summer and spring. The linkage between DMS and aerosol/clouds/CCNs is important, however, this topic is beyond the scope of our study. We used CTM simulations to compare with our observations.*

Most of the measurements are close to the surface and presumably within the atmospheric boundary layer (ABL), over ice covered seas, the MIZ and open water. Except for flights over ice covered seas in spring, the source of DMS near the surface is less interesting that the fate of that

DMS, since the source is most certainly local but the potential effects on aerosol and cloud dynamics may be far afield.

*The sampling altitudes are mentioned in table 1. Some samples were collected whithin the ABL. To show the boundary layer height, we referred to Aliabadi et al., (2016). Also, we added ice fraction (Figs S1, S3) to show the ice coverage during field campaigns.*

I'm not sure section 3.1 provides much insight. It's not clear what 'cloud processing of DMS' means, and correlations of DMS with water vapor, CO and ozone are subject to such a wide variety of dynamic and chemical influences that they are difficult to interpret. Conclusions here are vague. Fig. S4 merely reveals that DMS emission peaks in spring and water vapor peaks in summer, which is expected based on existing information.

*Agreed, we removed section 3.1 and Figs 5, S1, S2 and S4, and instead we added:*

[revised manuscript text omitted]

The vertical distribution of all these species is much more interesting. The authors are using an advanced CTM to simulate vertical distribution and transport. I'd like to see model results for H2O, CO and O3 plotted with the observations on Fig. S3, in a manner similar to that shown for DMS in Fig.6.

*Figure S2 shows vertical distribution for measurement and modeling simulations.*

Is there a well-defined atmospheric boundary layer on each flight? At what height? Does the profile of potential temperature, water vapor or vertical wind velocity indicate a well-mixed ABL and strong inversion? Is there evidence of atmospheric stability or of convection and mixing into the free troposphere? What does the CTM assume or simulate for vertical mixing on the flight dates and how does that compare with the observed profiles? How do profiles differ between open water, MIZ and ice covered regions? On p.10 the authors talk about limited vertical mixing and atmospheric stability during the April flights, but present no data or analysis to support this statement.

*Correct, we added more information:*

(Page 11, Line 4) DMS (g) vertical profiles are sensitive to the boundary layer height. For the summertime, Arctic the boundary layer height on various days (275±164 m), for the July 2014 campaign, is reported in Aliabadi et al. (2016). They showed that the profiles of the potential temperature exhibited a positive vertical gradient throughout the aircraft campaign (their Fig. 4). In addition, using vertical profiles of wind speed, they derived a positive gradient Richardson number (Ri) with a median of 2.5 (Their Fig. 7) throughout the aircraft campaign. The

magnitude of the positive gradient Richardson number is an indicator of the strength of thermal stability in the atmospheric boundary layer. Due to the strong thermally stable conditions during the field campaign, mixing was weaker compared to well-mixed boundary layers at mid latitudes. As a result the summertime measurements show a strong decrease in DMS(g) above the boundary layer. Although there is no reference for the April 2015 campaign boundary layer, we expect similar boundary layer characteristics in the stable Arctic boundary layer at high latitudes due to the even more reduced thermal forcing with large sun angles in the month of April compared to the month of July. The springtime measurements show a more uniform vertical profile suggesting transport in the free troposphere from open water sources that were relatively farther distance from the observation point in springtime than in summer.

*Also, please see:*

In all studies of atmospheric DMS over the ocean in the absence of deep convection, the concentration of DMS is elevated and well-mixed below the ABL inversion, dropping to near zero above the inversion. This is due to limited vertical transport and a relatively short photochemical lifetime. If NETCARE results differ from the usual DMS vertical distribution over the ocean, that's an interesting finding, implying convective transport into the free troposphere and perhaps an extended photochemical lifetime due to reduced water vapor or limited sunlight, all of which can be diagnosed with the CTM. In this regard, the vertical profile results for DMS in Section 3.2 are quite interesting.

From Fig.4a it looks like atmospheric DMS is often elevated and highly variable near the surface in mid-summer. At 3000m the concentrations are near zero and I presume this represents the background free troposphere (are these measurements at the detection limit?). But, the few measurements on Jul12/17 at 1000m are significantly elevated. Are these within a deep ABL or are they evidence of long range transport in the FT? The latter would be very interesting.

*The detection limit is ~ 7 pptv (Page 6, line 10). The measurement at 3000 m was below detection limit during July (mentioned in Table 1). For July 17, again the measurement was below 7 pptv at ~1000 m. However, for July 12th, simulation suggested a local (Lancaster Sound) influence:*

(Page 9, line 21) However, relatively high DMS mixing ratios (> 15 pptv) were observed for July 12th at high altitudes (> 800 m), and FLEXPART results shows influence of local source, Lancaster Sound for that day (mentioned in Section 4.2). On this day, NETCARE results do not follow the usual DMS vertical pattern of high DMS at the surface declining with altitude to near zero above the MBL. Instead, high concentrations aloft on July 12 imply convective transport into the free troposphere and potentially an extended photochemical lifetime due to reduced water vapor or limited sunlight.

During April, DMS(g) samples were collected above ice and snow surfaces, and heat fluxes were negligible. Figure S3 shows the ice fraction during the April 2015 campaign.

Fig. 7 shows CTM back trajectories for the 0-200m level, so they presumably apply to aircraft measurements at the lowest altitude only. Are the colors in Figs. 7/8 meant to signify surface source regions for low-level air sampled by the aircraft? I don't understand what is meant my 'air mass residence time in seconds before arriving at the aircraft location'. The authors should define 'potential emission sensitivity'. Also, the blue line indicating the flight path is not visible and probably too small properly plot on a map of this scale. Maybe it's best to just put a dot on the map indicating the flight location. Are the authors suggesting DMS measured near the surface over the MIZ and open water in mid-summer near Devon Island originated as far away as Hudson Bay? That doesn't make sense to me. So I'm not sure about the significance of Figs. 7 and 8. It would be more interesting if these were forward trajectories rather than back trajectories.

For July flights over the ice edge and open water in mid-summer (a time of extended daylight hours and perhaps greater photochemical oxidation) the author's conclusion that DMS sampled near the surface represents local emissions is quite reasonable. DMS within the ABL at 0-200m is likely to have originated from surface emissions quite near the sampling site under these conditions. I would be much more interested in back trajectories for measurements at the 1000m level, especially if this is above the ABL. If not, then the statement on p.9 that 'The decline in DMS mixing ratios with height may be due to a combination of weak vertical mixing and

photochemical reactions' is entirely comparable to conditions that exist over the ocean in most other locations in the world. This seems to be supported by the CTM results in Fig. 6. I would think they could provide evidence of vertical structure in the atmosphere to further support this conclusion.

*We changed Figs 7 and 8. New figures show back-trajectories for 1000 m.*

*Please note that we are interested to know the potential source of DMS, and as a result, we plot back-trajectories. More information is added to the manuscript to address your comments regarding the FLEXPART simulation:*

(Page 14, line 2) FLEXPART-ECMWF modeling was used to explore the origin of air samples measured along the Polar 6 flight tracks. Figures 6 and 7 show the potential source regions of these air samples four days before the releases along the flight path. More specifically, the response function is shown to all releases of a passive tracer, which in this case has properties of dry air. If this response function would be folded with an emission flux of the tracer the concentration of this tracer at the release location along, the flight paths could be calculated. We chose to show the potential emission sensitivity after four days. Sharma et al., (1999) showed that atmospheric DMS(g) lifetime was 2.5 to 8 days in the high Arctic. More details about FLEXPART and the potential emissions sensitivity (PES) could be found in Stohl et al. (2005) and references therein.

Figure 6 shows two examples of FLEXPART-ECMWF PES for 4-day back trajectories in July 2014: an influence from a broad area and especially Lancaster Sound (local region) and north on July 12[th] (Figure 6, left panel), and Hudson Bay, and Baffin Bay (south) on July 19[th] (Figure 6, right panel). A more detailed analysis of PES reveals that the measured air mass descended from >1500 m on July 19[th], which may explain the low DMS(g) mixing ratios.

Figure 7 shows some examples of FLEXPART-ECMWF PES simulations for 4-day back trajectories during April 2015. For the flights near Alert and Eureka on April 9 and 11, some DMS may have originated from ice-free areas of the Nares Strait and Baffin Bay (Figure 7, upper left and right panels, respectively). For the April 13 flight, the Norwegian Sea, North Atlantic Ocean and Hudson Bay are additional potential source regions (Figure 7, lower left panel). The highest

DMS, measured on April 20 near Inuvik is associated with the North Pacific Ocean (Figure 7, lower right panel).

Assuming a DMS atmospheric lifetime of 1 to 4 days, these results suggest that the DMS(g) measured during July 2014 originated primarily from the local region over Baffin Bay and the Canadian Arctic Archipelago. For spring 2015, the DMS(g) sampled was from a range of sources, including Baffin Bay, possibly the Norwegian Sea, the North Atlantic Ocean and the North Pacific Ocean.

For April flights, the measurements indicate high DMS concentrations near the surface, which is understandable given algal blooms associated with the spring melt, but also elevated concentrations aloft, which is very interesting. It would be good to know the fractional ice coverage for the April flights and if the surface exhibited extensive melting with open leads. We also need to see an analysis of the vertical structure of the atmosphere for these flights. Fig. 8 should show back trajectories for the upper flight levels to help diagnose the significance of long range transport and potential source regions. This seems more relevant than the source of surface atmospheric DMS, which is likely to be local and may be inhibited by atmospheric stability from contributing CCN and cloud dynamics aloft. Though it would be interesting to see a forward model trajectory for the surface air and an analysis of the potential to influence to aerosol aloft over remote, ice covered regions. Heat fluxes in the MIZ, where open water is present, might drive sufficient convection to transport surface emissions from the MIZ and open water to ice-covered regions, where cloud formation is driven by mixing from above. Hopefully, the CTM can reproduce these processes.

*The ice fraction is shown on Fig S3 for April 2015. Please see:*

(Page 9, line 27) During April, DMS(g) samples were collected above ice and snow surfaces, and heat fluxes were negligible. Figure S3 shows the ice fraction during the April 2015 campaign.

**Referee 2:**

First, I think the description of the measurement technique needs some clarification, even though it is based on methods described 20 years ago. My main questions to be cleared up: 1) Sample times range from 2 – 11 minutes. What is the flow rate, and what are the resulting sample volumes? 2) Reference to Sharma et al. (1999) as a description of the method seems incorrect as listed. The listed Sharma et al. (1999) paper describes a 10 L sample collected on molecular sieve. The reference Sharma et al. (1997) is a typo and should also be 1999. This appears to be the correct reference to the Tenax method. Sharma (1997) is the MSc thesis, which should contain all of the details but is not easily available which is why some extra detail is needed. The Sharma et al. (1999) paper lists the detection limit as -+6 pptv in a 2 L sample (defined as 2:1 S/N), and accuracy and precision of -+12% (not sure what this means, though). This manuscript refers to a -+ 12 pptv uncertainty (not specified if accuracy or precision). Surely this can't be constant for samples of different volumes. Was this determined based on the triplicate standard measurements, or was this somehow referenced back to Sharma et al. (1999)? Please clarify this, esp. since many of the samples reported seem to be in this range. 4) Sample storage tests: how much standard was loaded onto the cartridge? Were different loadings tested? Please also note the axis of the graph in Figure 3. Is this really the DMS mixing ratio in pptv, or some ratio to initial DMS addition? Is the uncertainty in the stability also considered in assessing the overall uncertainty of -+12 pptv?

*1000 mL of samples were collected in 5 mins with a flow rate of approximately 200 mL/min. For few samples the sampling time was shorter or longer than 5 mins, leading to different volume of samples. The uncertainties of DMS mixing ratio were 2 and 3 pptv for the minimum (400 mL) and maximum (2200 mL) of volumes, respectively.*
*We addressed these comments:*

(Page 6, line 1) Sampling collection time was $300 \pm 5$ seconds with a flow rate of $200 \pm 20$ mL/min (For few samples the sampling time was shorter or longer than 300 seconds, leading to different volume of samples).

A glass gas chromatograph (GC) inlet liner was used to pack 170±2 mg of Tenax. The Tenax packed in glass tubes was cleaned by heating to 200°C in an oven with a constant He flow of around 15 mL/min for 5 hours. The DMS samples were analyzed with using a Hewlett Packard 5890 gas chromatograph (GC) fitted with a Sievers Model 355 sulfur chemiluminescence detector (SCD). Two DMS(g) certified standards (1 and 50 ppmv) were used to calibrate the GC-SCD and to determine accuracy of the measurements by checking the standards against each other (for example, 1 microliter of 50 ppmv vs 50 microliters of 1 ppmv). Collection and analysis of samples were based on methods described by Sharma (1997), Sharma et al. (1999) and Rempillo et al. (2011). Uncertainty in the measurements was determined based on the standard deviation ($\sigma$) of DMS(g) standards and was ±12 pptv. The detection limit for this method is approximately 7 pptv.

Additional tests were performed to determine if there was significant loss of DMS(g) over time after collection. An experiment was performed to determine how long Tenax is able to store DMS(g) with no significant loss of concentration. This experiment was conducted in triplicate by loading of 50 µL of 1 ppmv DMS(g) standard and storing in a freezer at -25°C. In general, Tenax storage tests at -25°C showed that DMS losses were approximately 5% and 15% after 10 and 20 days respectively (Figure 3). The DMS(g) mixing ratios summarized in Table 1 are adjusted according to the result of this test.

P7, L.22. The GEOS-CHEM model is certainly widely used for many applications, but as I looked at the Mungall et al., 2016 paper, it seemed that there were significant issues with the DMS emissions specific for the region that was studied, and these conclusions suggest that the GEOS CHEM model needs to be applied with caution for specific locations and seasons for dealing with a highly variable compound such as DMS. Just because it was used is not necessarily an endorsement for its applicability for these measurements, at least without some caveats.

P8, L6. Meteorology and profiles. It is unusual to see the vertical profiles of species averaged for an entire campaign. The DMS measurements need to be related to the atmospheric vertical structure and variability during each flight, or there should be some demonstration that a more suitable analysis is found by taking mission averages.

*As global models often track measurement features but may be slightly off in location or altitude, the model-measurement comparison is more robust if we consider mission averages, particularly in this case where we have limited DMS measurements. We added text on page 12 line 14 to indicate this rationale for the model-measurement comparisons.*

*The $O_3$, $H_2O$ and CO vertical distribution for DMS sampling days are shown in figure S2 and suggest significant uniformity throughout the sampling campaign. We added the uncertainty. The plots are used to compare $O_3$, $H_2O$ and CO during summer and spring. In addition, the O3 depletion events are reported in Table 1.*

P8, L 16. Please provide the reference(s) to the heterogeneous oxidation of DMS on or within aerosols.

P8, L 24. While sources of DMS and H2O are the ocean (though there could be other sources of DMS), different correlations could be observed due to the high spatial and temporal variability of DMS (compared to H2O) and effect of temperature on either gas. I assume this discussion of the correlation is related to the argument about the greater presence of clouds in spring vs. summer, and (again) the assertion of a major role of cloud processing of DMS. The presence (note typo on P8, L25) of clouds could potentially have a greater impact on OH production below or above the clouds with subsequent impact on DMS lifetime. Perhaps this was suggested by the statement regarding clouds during transport, but this seems a speculation that could be checked from observations. It is not clear what the authors are saying regarding the effect of clouds: : :.reduction of DMS due to heterogeneous O3 oxidation, or increase of DMS due to slower OH oxidation? Unfortunately, the data is not available to support either impact so this just remains as speculation.

*Correct, the study of aerosol/clouds/CCNs is beyond the scope of this study. Please note that Section 3.1 has been removed.*

P9, L 25. The authors reference again the work reported by Mungall et al., 2016 who measured DMS at the same general location and time period (as far as I can tell) as the airborne data for July,

2014. It would be helpful to discuss the airborne measurements and the impact of the very large variability in DMS emissions that were found in Mungall et al. Perhaps there were times of reasonable overlap between the ship based and airborne measurements that could also be compared. It seems that the large DMS mixing ratios observed during parts of the Mungall et al. cruise were not sampled by the aircraft. Furthermore, the reported range of a factor of 60 in the calculated DMS flux should further raise a cautionary flag about how well climatological models and emission estimates might be applied to the data.

*The Amundsen and Polar 6 measurements occurred during July 2014, but not exactly in the same time and location. We added more information and compared the event during 18-20 July:*

(Page 9, line 5) Mungall et al. (2016) also suggested LRT of DMS from marine regions outside Baffin Bay and Lancaster Sound area, and observed an episode of elevated DMS (g) mixing ratios with values of 400 pptv or above occurred on 18–20 of July. The airborne measurement, showed decline of DMS (g) mixing ratios by height during July 17, and relatively low DMS mixing ratios during July 19th and 20th (see Table 1).

P10, L 19. Here and elsewhere in the manuscript the possibility of long-range transport of DMS is mentioned. In this context, it would be helpful if the authors could provide some reasonable estimate of the lifetime of DMS and the possible transport times that are considered "long-range" vs. local.

P13, L9. Could the authors make some estimate, based on typical DMS lifetimes, what the source region DMS mixing ratios would need to be to support the observed mixing ratios in spring?

*The life time is assumed to be less than 4 days, and we referred to Sharma et al. (1999):*

(Page 14, line 7) We chose to show the potential emission sensitivity after four days. Sharma et al., (1999) showed that atmospheric DMS(g) lifetime was 2.5 to 8 days in the high Arctic. More details about FLEXPART and the potential emissions sensitivity (PES) could be found in Stohl et al. (2005) and references therein.

Figure 6 shows two examples of FLEXPART-ECMWF PES for 4-day back trajectories in July 2014: an influence from a broad area and especially Lancaster Sound (local region) and north on July 12[th] (Figure 6, left panel), and Hudson Bay, and Baffin Bay (south) on July 19[th] (Figure 6, right panel). A more detailed analysis of PES reveals that the measured air mass descended from >1500 m on July 19[th], which may explain the low DMS(g) mixing ratios.

Figure 7 shows some examples of FLEXPART-ECMWF PES simulations for 4-day back trajectories during April 2015. For the flights near Alert and Eureka on April 9 and 11, some DMS may have originated from ice-free areas of the Nares Strait and Baffin Bay (Figure 7, upper left and right panels, respectively). For the April 13 flight, the Norwegian Sea, North Atlantic Ocean and Hudson Bay are additional potential source regions (Figure 7, lower left panel). The highest DMS, measured on April 20 near Inuvik is associated with the North Pacific Ocean (Figure 7, lower right panel).

Assuming a DMS atmospheric lifetime of 1 to 4 days, these results suggest that the DMS(g) measured during July 2014 originated primarily from the local region over Baffin Bay and the Canadian Arctic Archipelago. For spring 2015, the DMS(g) sampled was from a range of sources, including Baffin Bay, possibly the Norwegian Sea, the North Atlantic Ocean and the North Pacific Ocean.

P10, L 25. I am not a fan of averaged vertical profiles for variable species (such as aerosols) without some indication of the variability of the measurements over the different profiles. I assume that the higher particle number is associated with the specific profile in July where the measured DMS is about 100 ppt. Or is the higher level of aerosol more widespread? Also, the springtime levels of DMS are even higher than those found in summer, but no significant aerosol production here? This level of discussion and interpretation is incomplete and not particularly informative.

P11, L 3. Thin aerosol layers are common in the Arctic spring. It seems very unlikely that the high altitude peak in aerosol is in any way related to DMS. If the authors really consider the DMS source likely, perhaps some analysis of what sort of conditions might be necessary to produce the observed aerosol abundance.

*Agreed, we removed Fig 5 and discussion about aerosol (study of aerosol is not the focus of this manuscript).*

P11, L 13. Do I understand correctly that the GEOS-CHEM results shown in Figure 6 are compared only for the 2 – 10 minute sample integration time of the measurements? It would be interesting to see some examples of the variability of GEOS CHEM results to perhaps provide a more detailed context for the very limited number of actual samples that were collected. Often models track measurement features, but may be slightly off in time or altitude.

*Correct, We added more information:*

==(Page 12, line 13) Caution should be used in interpreting the model-measurement comparisons since these comparisons are conducted over a very limited number of measurement periods and the spatial and temporal resolution of these measurements is a challenge for a global model to simulate.==

P11, L 23. I am not sure what the authors want to convey when they claim that the GEOS CHEM data and the measurements are "within their respective uncertainties". Perhaps "variability" is a better term than uncertainties in this case. There are clearly more uncertainties in the model results beyond just the calculated mixing ratios.

*Yes- We changed "uncertainty" to "variability".*

P11, L25. From Figure 6, the variability with altitude actually seems less in the summer compared to spring, though the gradient is less in the spring, as noted. Summer range about 20 – 40 ppt at low altitudes, <10 ppt at max altitude; Spring 30 – >50 ppt at all altitudes. Could the authors please quantify the difference in seasonal variability at different altitudes to demonstrate their assertion?

*Thank you- We corrected the text:*

(Page 13, line 2) Both the simulated and measured DMS(g) profiles during spring (~ 30 to >50 pptv) show more variability at all altitude below 4 km than in summer (~ 20 to 40 pptv at low altitudes and <10 pptv at higher altitudes).

P12, L16. 30% difference might also be due to incorrect distribution and intensity of model DMS emissions.

*A cautionary statement is added to the manuscript:*

(Page 13, line 11) The monthly mean seawater DMS field used in our simulations is based on very limited observations from this region (Lana et al., 2011). Datasets of seawater DMS with higher spatial and temporal resolution are needed but are still under development.

P13, L5. I am concerned about the conclusions from the Flexpart analysis where the authors claim a large influence for specific regions. As noted, these analyses suggest where the sampled air spent time in the boundary layer. This analysis must be combined with an emission distribution to provide the source attribution. For example, without additional information, it is misleading to suggest that North Pacific was a significant source of DMS measured in the Arctic. It is a potential source. If the authors have some other data (ocean color, for example) to indicate DMS emission in the North Pacific at that time, then that could be a basis for the claim. Without this emission information, the Flexpart analysis can only indicate "potential" regions.

*Correct: We added word "potential" and more information about FLEXPART to the text:*

(Page 14, line 2) FLEXPART-ECMWF modeling was used to explore the origin of air samples measured along the Polar 6 flight tracks. Figures 6 and 7 show the potential source regions of these air samples four days before the releases along the flight path. More specifically, the response function is shown to all releases of a passive tracer, which in this case has properties of dry air. If

this response function would be folded with an emission flux of the tracer the concentration of this tracer at the release location along, the flight paths could be calculated. We chose to show the potential emission sensitivity after four days. Sharma et al., (1999) showed that atmospheric DMS(g) lifetime was 2.5 to 8 days in the high Arctic. More details about FLEXPART and the potential emissions sensitivity (PES) could be found in Stohl et al. (2005) and references therein.

Figure 6 shows two examples of FLEXPART-ECMWF PES for 4-day back trajectories in July 2014: an influence from a broad area and especially Lancaster Sound (local region) and north on July 12[th] (Figure 6, left panel), and Hudson Bay, and Baffin Bay (south) on July 19[th] (Figure 6, right panel). A more detailed analysis of PES reveals that the measured air mass descended from >1500 m on July 19[th], which may explain the low DMS(g) mixing ratios.

Figure 7 shows some examples of FLEXPART-ECMWF PES simulations for 4-day back trajectories during April 2015. For the flights near Alert and Eureka on April 9 and 11, some DMS may have originated from ice-free areas of the Nares Strait and Baffin Bay (Figure 7, upper left and right panels, respectively). For the April 13 flight, the Norwegian Sea, North Atlantic Ocean and Hudson Bay are additional potential source regions (Figure 7, lower left panel). The highest DMS, measured on April 20 near Inuvik is associated with the North Pacific Ocean (Figure 7, lower right panel).

Assuming a DMS atmospheric lifetime of 1 to 4 days, these results suggest that the DMS(g) measured during July 2014 originated primarily from the local region over Baffin Bay and the Canadian Arctic Archipelago. For spring 2015, the DMS(g) sampled was from a range of sources, including Baffin Bay, possibly the Norwegian Sea, the North Atlantic Ocean and the North Pacific Ocean.

P13, L16. The link established in this data set between aerosol formation and DMS is very weak and essentially presented as a given with no detailed analysis. While we all seek global significance of atmospheric chemistry and a potential link to climate, I think that it is an overreach to state that, without much greater data coverage and analysis, the observed vertical variation will significantly impact the Earth's radiation budget.

*Agreed, the discussion of aerosol is removed.*

Figures 1 and 2. These are not particularly helpful figures to identify the sample locations. I would suggest combining each season's flights into several composites that showed 1) a map of flight tracks, 2) an altitude-longitude (and/or latitude) cross-section that identified the location of the samples and the flight tracks. Given only a few flights, a color code could make things relatively easy to follow.

*Figs 1 and 2 show the sampling locations and altitudes. To address this comment, we added more details about the sampling location (Lat/Lon/Alt) in Table 1.*

Figure 3. Already noted.Vertical axis mislabeled?

*The vertical axis shows the DMS mixing ratios, and the mixing ratio of DMS standard used for this experiment is 1 pptv.*

Figure 5. Possible to add shading to each mean profile to show range or std deviation in each bin?

*Fig 5 is removed.*

Figure 7. I could not distinguish the blue lines for the flight tracks. Are these calculated for a single sample or some altitude? Please specify. I certainly do not understand the units for potential emission sensitivities. I thought that these indicated the time spent in the boundary layer (in this case 0 – 200 m) during the period of the back trajectories (4 days here). Is there a factor of 1000 missing here?

*Please note that we added more information about FLEXPART and changed figures 7 and 8.*

Table 1. It would be helpful to add the flight number and altitude to this table.

*Agreed, we added more information to Table 1.*

**Referee 3:**

The overall methodology is probably good but the description of the DMS measurements is confusing and would benefit from rewriting and reordering some sentences. Why did the sample collection times vary so much? Is there a reason for this? It seems like different collection times will result in different amounts of sample collected resulting in different limits of detection. Please comment on this and clarify. The paragraph beginning on page 5 line 23 is particularly confusing. It seems that this paragraph was intended to describe the calibration methodology but this is not obvious. It is stated that "Three Tenax tubes were injected with standard DMS along with one blank Tenax tube for each test period: : :", Why? What is the meaning of this? This is followed by a statement about calibrating the GC-SCD with 1 and 50 ppmv gas DMS standards.

Where did the laboratory get these standards. Were they certified standards etc. Collection and analysis were referenced to Sharma and Rempillo after the collection was already briefly described prior to this statement. It is stated that the uncertainty is 12% with this method but is that somehow independent of the amount of sample collected AND the mixing ratio of the sample that was collected? Please clarify and add a brief description of the Sharma and Rempillo methods and how the 12% uncertainty is determined.

The Tenax storage test shown in Figure 3 needs further discussion. The authors prepared a 1 pptv sample which is impressive. Would like details on how they did that. It is not clear what the standard deviation for each test represents. How many times were the samples analyzed etc.? And does the test have meaning given the uncertainty in the measurements? What is the LOD of the measurements?

*Thank you- we tried to address these comments:*

*1000 mL of samples were collected in 5 mins with a flow rate of approximately 200 mL/min. For few samples the sampling time was shorter or longer than 5 mins, leading to different volume of samples. The uncertainties of DMS mixing ratio were 2 and 3 pptv for the minimum (400 mL) and maximum (2200 mL) of volumes, respectively.*

(Page 6, line 1) Sampling collection time was $300 \pm 5$ seconds with a flow rate of $200 \pm 20$ mL/min (for few samples the sampling time was shorter or longer than 300 seconds, leading to different volume of samples).

A glass gas chromatograph (GC) inlet liner was used to pack 170±2 mg of Tenax. The Tenax packed in glass tubes was cleaned by heating to 200°C in an oven with a constant He flow of around 15 mL/min for 5 hours. The DMS samples were analyzed with using a Hewlett Packard 5890 gas chromatograph (GC) fitted with a Sievers Model 355 sulfur chemiluminescence detector (SCD). Two DMS(g) certified standards (1 and 50 ppmv) were used to calibrate the GC-SCD and to determine accuracy of the measurements by checking the standards against each other (for example, 1 microliter of 50 ppmv vs 50 microliters of 1 ppmv). Collection and analysis of samples were based on methods described by Sharma (1997), Sharma et al. (1999) and Rempillo et al. (2011). Uncertainty in the measurements was determined based on the standard deviation (σ) of DMS(g) standards and was ±12 pptv. The detection limit for this method is approximately 7 pptv.

DMS measurements and discussion – the decline in DMS mixing ratios with height in July is essentially what is expected and the pattern been seen in a number of previous studies. As stated, it results from primarily from fast photochemical destruction in the absence of deep convection as the lifetime of DMS is fairly short in July (_ _ 1 day). The data points above the surface (1 and 3 km) could be interesting but it would be instructive to know/understand the confidence that the authors have in these measurements with respect to LODs etc.

*The detection limit is ~ 7 pptv (Page 6, line 10). The measurement at 3000 m was below detection limit during July (we mentioned in the Table 1). For July 17, again the measurement was below 7 pptv at ~1000 m. However, for July 12$^{th}$, simulation suggested a local (Lancaster Sound) influence:*

(Page 9, line 21) However, relatively high DMS mixing ratios (> 15 pptv) were observed for July 12$^{th}$ at high altitudes (> 800 m), and FLEXPART results shows influence of local source, Lancaster Sound for that day (mentioned in Section 4.2). On this day, NETCARE results do not follow the usual DMS vertical pattern of high DMS at the surface declining with altitude to near zero above the MBL. Instead, high concentrations aloft on July 12 imply convective transport into the free troposphere and potentially an extended photochemical lifetime due to reduced water vapor or limited sunlight.

Also related to that, I agree with other reviewers that the authors should include more discussion of the vertical structure of the atmosphere in section 3.1 It is important to know if there evidence of atmospheric stability or of convection and mixing into the free troposphere.

*More information is added:*

(Page 11, line 4) DMS (g) vertical profiles are sensitive to the boundary layer height. For the summertime, Arctic the boundary layer height on various days (275±164 m), for the July 2014 campaign, is reported in Aliabadi et al. (2016). They showed that the profiles of the potential temperature exhibited a positive vertical gradient throughout the aircraft campaign (their Fig. 4). In addition, using vertical profiles of wind speed, they derived a positive gradient Richardson number (Ri) with a median of 2.5 (Their Fig. 7) throughout the aircraft campaign. The magnitude of the positive gradient Richardson number is an indicator of the strength of thermal stability in the atmospheric boundary layer. Due to the strong thermally stable conditions during the field campaign, mixing was weaker compared to well-mixed boundary layers at mid latitudes. As a result the summertime measurements show a strong decrease in DMS(g) above the boundary layer. Although there is no reference for the April 2015 campaign boundary layer, we expect similar boundary layer characteristics in the stable Arctic boundary layer at high latitudes due to the even more reduced thermal forcing with large sun angles in the month of April compared to the month of July. The springtime measurements show a more uniform vertical profile suggesting

transport in the free troposphere from open water sources that were relatively farther distance from the observation point in springtime than in summer.

The April results are definitely quite interesting. I am surprised at both the surface measurement and aloft mixing ratios since this time of year is early for substantial biological productivity I would think. The authors didn't mention previous ob servations from the NASA DC-8 during ARCTAS (https://www-air.larc.nasa.gov/cgibin/ ArcView/arctas). The results in paper contrast with the ARCTAS data in spring where lower DMS mixing ratios were observed (below detection limit to a few pptv and a max of 1 pptv in the free troposphere). It would be interesting to describe leads observed etc. in the region of sampling.

*During NETCARE campaign, there were not leads in the sampling locations. We added more information about ARCTAS study:*

(Page 4, line 12) Observations of the NASA DC-8 during ARCTAS (https://www-air.larc.nasa.gov/cgibin/ArcView/arctas) showed low DMS mixing ratios in spring (below detection limit to a few pptv in the boundary layer and a maximum of 1 pptv in the free troposphere) (Simpson et al., 2010; Lathem et al., 2013).

I am curious about the DMS emission source inventories used in the model and where
these came from during springtime.

*The DMS emissions inventory used in the model are from Lana et al. (2011) monthly mean DMS climatology, which includes both the springtime and summer.*

Minor things:

P3 line 8 – describe CLAW hypothesis

(Page 3, line 9) Charlson et al. (1987) hypothesized that DMS could provide a negative feedback to stabilize the global warming (CLAW hypothesis). Although no evidence in support of the hypothesis has been found (Quinn and Bates, 2011), DMS(g) emissions may play an important role in the climate of remote areas with low aerosol concentrations, such as in the Arctic (Carslaw et al., 2013, Leaitch et al., 2013, Levasseur 2013, Croft et al., 2016a).

P4 line 13 add s to altitudes to make it plural

*Thank you, We made the change.*

P4 line 20 – suggest replacing "act" with "appear"

*Thank you, We made the change.*

P5: As suggested above rewrite paragraph beginning on line 17

*Thank you, We made the change.*

P8 line 25 replace "higher present" with "a higher presence"

*Section 3.1 is removed.*

P8 line 26 – make Cloud plural – "Clouds"

*Thank you, We made the change.*

P10 line 5 eliminate comma after mixing ratios

*Thank you, We made the change.*

---

## Author Response (AR2)

Dear Prof. Russell,

We are pleased to resubmit the revised version of MS# ACP-2017-33: *"Boundary layer and free tropospheric dimethyl sulfide in the Arctic Spring and Summer".* We thank you and the reviewers for your time and valuable comments.

We believe that we have addressed all the concerns. Please find comments, our responses (in italics) and the modified text in the manuscript (highlighted) as outlined below.

Your Sincerely,

Roghayeh Ghahremaninezhad (Roya)

Postdoctoral Fellow, Air Quality Research Division,

Environment and Climate Change of Canada,

Toronto, ON, Canada

Tel: +1 416 739 4690

Email: Roghayeh.Ghahremaninezhad@Canada.ca

Report #1

P5 line 23: authors sate that mass flow was controlled at 200 ± 20 mL/min. They restate that on the following page P. 6 line 1 – redundant

*Thank you, We removed the extra information.*

(Page 6, line 3) Sampling collection time was 300 ± 5 seconds (for few samples the sampling time was shorter or longer than 300 seconds, leading to different volume of samples).

P9 line 22…FLEXPART results shows influence from a local source, Lancaster Sound, for…P11line 4: For the summertime, Arctic…rewrite this sentence. Perhaps: For the July 2014 campaign, Aliabadi et al. (2016) report an average boundary layer height of 275 ± 164 m. P11 line 17…open water sources that were a relatively farther distance… P14 line 6: an emission flux of the tracer, the concentration of this tracer at the release location along the flight…

*Thank you, We made the changes.*

(Page 9, line 25) and FLEXPART results shows influence from a local source, Lancaster Sound (Page 11, line 9) For the July 2014 campaign, Aliabadi et al. (2016) report an average boundary layer height of 275±164 m.

(Page 11, line 21) from open water sources that were a relatively farther distance from the observation point in springtime than in summer.

(Page 14, line 9) an emission flux of the tracer, the concentration of this tracer at the release location along the flight paths…

Report #2:

I agree with the previous reviewers that the authors need to state where they got their certified standards (1 and 50 ppmv gas DMS standards).

*The certified DMS(g) standards (1 and 50 ppmv) were provided by "Praxair".*

(Page 6, line 8) Two DMS(g) certified standards (1 and 50 ppmv)…

They also need to clarify what is meant by the uncertainty of the measurements was determined by "the standard deviation of ±12% of DMS standards". Do they mean that they ran the standards x number of times and the precision of their analysis of their standards was ± 12%? They should re-phrase this and state what they did. More importantly though, that does not represent the uncertainty in the measurements. One factor among many in the uncertainty of their measurements is the uncertainty in their standards etc. They should state what this is (i.e. 1 ppmv ± x). The uncertainty in their measurements in terms of % uncertainty will certainly go up as they approach their LOD. I am not asking them to do a full analysis of the uncertainty in their measurements but at least acknowledge what they have stated is the precision in replicate analyses of their standard. (if that is indeed what they have stated).

*Thank you, We addressed this comment.*

Precision of analysis was ±12 pptv and was determined based on the standard deviation (σ) of triplicate measurements of DMS(g) standards.

Figure 3 still does not make sense to me. If their LOD is as stated 7 pptv, how can they measure fractions of a pptv? Perhaps they mean ppmv? (or perhaps I am missing something?)

*Correct, That was a typing mistake. Now, the unit for DMS mixing ratio is ppmv in figure 3.*

Report #3

This manuscript summarizes 38 atmospheric DMS samples collected on 11 flights during two flight campaigns. I do not think there are sufficient samples to claim "DMS profiles". Perhaps it can be summarized as boundary layer and free tropospheric concentrations of DMS in the Arctic.

*We changed the title of the manuscript to:* ==*Boundary layer and free tropospheric dimethyl sulfide in the Arctic Spring and Summer*==

*Also we added a sentence in the conclusion to address this point:*

(Page 15 line 8) ==This study includes very limited spatial and temporal extent, and further vertical profile measurements of Arctic DMS(g) is recommended.==

It would be useful to include the boundary layer height in Table 1.

*We added the boundary layer height (Aliabadi et al., 2016) to the Table 1.*

The authors use GEOS-CHEM and Flexpart to interpret the DMS data. GEOS-CHEM is problematic since the DMS flux is very patchy in the Arctic. Never the less it is interesting to see how the model compares with the limited data.

The authors have revised the manuscript based on three thorough reviews and have addressed the issues raised by the reviewers. I think the paper should be published taking into account the 2 suggestions above.

[revised manuscript text omitted]